# Regularization is Enough for Last-Iterate Convergence in Zero-Sum Games

## Abstract

Recent literature has witnessed a rising interest in learning Nash equilibrium with a guarantee of last-iterate convergence. In this paper, we introduce a novel approach called Regularized Follow-the-Regularized-Leader (RegFTRL) for the purpose of learning equilibria in two-player zero-sum games. RegFTRL is an efficient variant of FTRL, enriched with an adaptive regularization that encompasses the well-known entropy regularization as a special case. In the context of normal-form games (NFGs), our proposed RegFTRL algorithm exhibits the desirable property of last-iterate linear convergence towards an approximated equilibrium. Furthermore, it converges to an exact Nash equilibrium through adaptive adjustments of the regularization. In extensive-form games (EFGs), we demonstrate that the entropy-regularized Multiplicative Weights Update (MWU), a specific instance of RegFTRL, can achieve a last-iterate linear convergence rate towards the quantal response equilibrium, all without the need for either an optimistic update or reliance on uniqueness assumptions. These results show that regularization is enough for last-iterate convergence. Additionally, we propose FollowMu, a practical implementation of RegFTRL with a neural network as the function approximator, for model-free learning in sequential non-stationary environments. Finally, empirical results substantiate the theoretical properties of RegFTRL, and demonstrate that FollowMu achieves favorable performance in EFGs.

## 1 Introduction

Online learning has a rich history that is inextricably intertwined with the advancement of game theory, convex optimization, and machine learning. One of the earliest manifestations of online learning can be attributed to Brown (1949)'s proposal of fictitious play as a method for solving two-player zero-sum games. Ensuing result (Robinson, 1951) has revealed that iteratively computing a best response to each other's history of play in (zero-sum) matrix games leads to convergence to the set of Nash equilibria. This kind of learning paradigm can be linked to the notion of no-regret learning (Cesa-Bianchi & Lugosi, 2006), which shares a common historical thread with game theory that dates back to Blackwell's approachability theorem (Blackwell, 1956; Abernethy et al., 2011).

It is folklore that the time-average policies of no-regret algorithms in self-play converge to a Nash equilibrium in two-player zero-sum games (called *average-iterate convergence*) (Cesa-Bianchi & Lugosi, 2006). A plethora of online learning algorithms, including the celebrated Online Mirror Descent (OMD) (Warmuth et al., 1997) and Follow-the-Regularized-Leader (FTRL) (Abernethy et al., 2008), ensure that the worst case regret is upper bounded sublinearly with learning iterations (Shalev-Shwartz, 2012), thus allowing for a global on-average convergence to the Nash equilibrium over time. A myriad of studies have significantly expanded the applicability of the no-regret theorem to a broader class of settings, covering extensive-form games (EFGs) (Zinkevich et al., 2007; Hoda et al., 2010), Markov games (MGs) (Bai et al., 2020; Tian et al., 2021), differential games (i.e. smooth games) (Vlatakis-Gkaragkounis et al., 2019), and auctions (Deng et al., 2021).

However, the average-iterate convergence characteristic poses significant challenges in game theory and its practical applications, especially when representing policies using deep networks (Heinrich & Silver, 2016). In most game settings, averaging neural network weights does not directly correspond to an average of the policies represented by those networks (Heinrich et al., 2015; Daskalakis et al., 2018). To mimic agents' average behaviors, it is often required to maintain an additional reservoir

buffer, typically hundreds or even thousands of times the size of the game, to store past transition data (Heinrich & Silver, 2016) or historical network parameters (Lanctot et al., 2017), leading to extremely high memory demands. Moreover, the average policy cannot be represented precisely due to the inherent neural network approximation error.

Accordingly, it is imperative to develop no-regret algorithms that converge to (approximate) Nash equilibrium without averaging (called *last-iterate convergence*). However, previous researches have demonstrated that the standard no-regret algorithms can lead to cyclic behaviors (Zinkevich et al., 2005; Omidshafiei et al., 2019) or even chaotic behaviors (Sato et al., 2002) of the real-time policy. In recent literature, methods based on optimistic gradients (Daskalakis & Panageas, 2019; Wei et al., 2021), predictive updates (Yadav et al., 2018), and opponent learning awareness (Foerster et al., 2018) shed lights on how to break the cyclic behaviors by predicting opponents' next moves. Algorithmically, these methods can be considered variations of the optimistic/extra-gradient methods, where the gradient dynamics are modified through the introduction of approximate second-order information (Schäfer & Anandkumar, 2019). However, in different types of games, the second-order information may have different effects on the learning dynamics. The top plots in

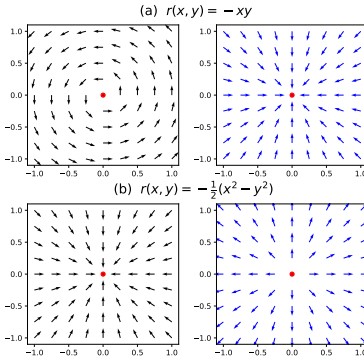

Figure 1: Learning dynamics (black arrows) and second-order dynamics (blue arrows) in the (A) Hamiltonian game, and (B) potential game.

Figure 1 depict the rotations of gradient dynamics around the equilibrium in a Hamiltonian game [1] where the second-order dynamics converge. This highlights the effectiveness of introducing (approximate) second-order information in jumping out of cycles. However, in some potential games (as depicted in the bottom plots), the use of second-order information can impede convergence, particularly when the real-time policy exhibits chaotic behavior.

The contributions of this paper are mainly three-fold: (1) instead of using the approximate second-order information, we introduce an extra regularization, independent of the game types, into the underlying game to enhance its potential component and thus establish general-case last-iterate convergence; (2) by incorporating the regularization, we present a variant of FTRL called **Regularized FTRL** (RegFTRL) that is able to converge at an exponentially fast rate in both NFGs and EFGs without either the optimistic update or the uniqueness assumptions, and investigate two approaches, annealing and adaption approaches, to build algorithms that converge to an exact Nash equilibrium; and (3) we propose a model-free reinforcement learning algorithm, **FollowMu**, as a practical implementation of RegFTRL, and validate its performance in Kuhn & Leduc and Phantom Tic-Tac-Toe.

## 1.1 RELATED WORK

Research in the realm of last-iterate convergence can be roughly divided into two lines: the optimistic update paradigm and the regularization technique. In the optimistic update approach, previous studies (Rakhlin & Sridharan, 2013; Daskalakis et al., 2018; Liang & Stokes, 2019; Mokhtari et al., 2020) have investigated the last-iterate convergence in simple unconstrained cases, which are not directly applicable to the NFG/EFG setting. In cases where a unique Nash equilibrium is assumed, Daskalakis & Panageas (2019) and Wei et al. (2021) have extended the scope of research by providing last-iterate convergence guarantees for Optimistic Multiplicative Weights Update (OMWU) (corresponds to Optimistic FTRL with an entropy regularizer) in NFGs, while Wei et al. (2021) further prove the convergence of Optimistic Gradient Descent/Ascent (OGDA) (corresponds to optimistic OMD with a $L_2$ regularizer) without the uniqueness assumption. In the context of EFGs, the pioneering work by Farina et al. (2019) empirically demonstrates the last-iterate convergence of OMWU, while Lee et al. (2021) subsequently establish theoretical proofs with the uniqueness assumption. Recently, Gorbunov et al. (2022); Cai et al. (2022) have extended the convergence properties of OGDA to monotone games, which include many common classes of games, such as zero-sum polymatrix games and concave-convex games. However, the optimistic update is not flawless. From a theoretical standpoint, OMWU still lacks an explicit last-iterate convergence rate,

---

[1]See Appendix C for discussions on Hamiltonian and potential games. In brief, an NFG is potential if there is a single potential function $g$ such that $V_{\pi^1,\pi^2} - V_{\hat{\pi}^1,\pi^2} = -g(\pi^1,\pi^2) + g(\hat{\pi}^1,\pi^2)$ for all $\pi^1, \hat{\pi}^1, \pi^2$.

even in NFGs, without the reliance on uniqueness assumption (Wei et al., 2021; Lee et al., 2021). Additionally, the analysis of OMWU in EFGs is built over the sequence-form strategy (Wei et al., 2021; Lee et al., 2021), which exhibits limitations in scaling to large games. In practical terms, the implementation of the optimistic update approach often necessitates the computation of multiple strategies at each iteration. Furthermore, OGDA has high per-iteration complexity due to the costly projection operations at each iteration, which adds to the computational burden. In contrast, our proposed approach, RegFTRL, offers distinct advantages. It obviates the requirement for the uniqueness condition and emphasizes the behavior-form strategy, making it more compatible with reinforcement learning and readily adaptable to large-scale games.

Within the realm of learning dynamics, regularization technique has emerged as pivotal tools for accelerating convergence (Pérolat et al., 2021; Abe et al., 2022; Sokota et al., 2023; Abe et al., 2023). Pérolat et al. (2021) conducted a comprehensive analysis of the impact of entropy regularization on continuous-time dynamics, and propose a reward transformation method to achieve linear convergence in EFGs using counterfactual values. However, it is imperative to note that their theoretical findings pertaining to continuous-time dynamics do not inherently extend to the desired discrete-time results. Moreover, the use of counterfactual values presents scalability challenges in large-scale settings (McAleer et al., 2023). Furthermore, their reward transformation technique can lead to estimation issues due to the arbitrarily cumulative sum. Wang et al. (2022) shows that the GDA algorithm, with a decreasing learning rate, achieves last-iterate convergence in strongly monotone games. In the context of monotone games, the establishment of strong monotonicity is achievable through the incorporation of a strongly convex regularizer. Abe et al. (2022) proposed Mutation-Driven MWU (M2WU), a variant of MWU for solving NFGs. M2WU incorporates an additional mutation term and, in essence, constitutes an instantiation of RegFTRL with moment projection serving as the regularization. Similar to our work, Magnetic Mirror Descent (MMD) (Sokota et al., 2023) and FTRL with Slingshot Perturbation (FTRL-SP) (Abe et al., 2023) investigate the influences of general-case regularization on last-iterate convergence, and both provide the linear convergence rate to the regularized equilibrium, albeit FTRL-SP with a more strict restriction on the learning rate. However, MMD exclusively achieves convergence towards an approximated equilibrium in NFGs, while the concurrent work FTRL-SP provides a convergence rate to an exact Nash equilibrium with the $L_2$ regularizer in monotone games. Furthermore, their analyses could not encompass the behavior-form EFGs considered in our paper. Due to the uniqueness of the quantal response equilibrium (QRE), certain endeavors have attempted to combine optimistic update with additional entropy-regularization to remove the uniqueness assumption associated with OMWU (Cen et al., 2021; Liu et al., 2022), but these approaches still inherit limitations of the optimistic update paradigm.

Compared with the aforementioned related works that either focus on matrix games (Abe et al., 2022; Sokota et al., 2023), only consider a special regularization (Abe et al., 2022), or assume continuous-time feedback (Pérolat et al., 2021), we go one step further and prove that RegFTRL converges to an exact Nash equilibrium via general-case regularization in NFGs, and converges exponential fast to the QRE in behavior-form EFGs without the continuous-time feedback assumption. Additionally, through empirical observations, we substantiate that RegFTRL equipped with alternative regularization, consistently exhibits last-iterate convergence in behavior-form EFGs. Collectively, our findings underscore that regularization is enough for the last-iterate convergence in zero-sum games.

## 2 PRELIMINARIES

### 2.1 EXTENSIVE-FORM GAMES

The representation of a two-player zero-sum EFG $\langle \mathcal{N}, \mathcal{S}, \mathcal{A}, \mathcal{I}, \mathcal{P}, \{r^h\}_{h=1}^H \rangle$ is based on a game tree of depth $H$, modeling the sequential interactions involving each player $i \in \mathcal{N} = \{1, 2\}$ and a chance player $c$. At each history $s \in \mathcal{S}$ at time $h \in [H]$, corresponding to a node at level $h$ in the finite rooted game tree, the player function $\mathcal{P}(s) \mapsto i \in \mathcal{N} \cup \{c\}$ determines a player or chance to play an action $a \in \mathcal{A}(s)$. As a result, player 1 will receive a reward $r^h(s, a) \in [0, 1]$ (and player 2 will receive a reward $-r^h(s, a)$), and the history will transition to its successor history $s' = sa$ at time $h + 1$. We denote $s' \sqsubset s$ if $s'$ is led from $s$. Due to the imperfect-information, at each history $s$,

only an **information state** $I \in \mathcal{I}$ can be observed, where histories $s \in I$ are indistinguishable for the current player. We use $I(s)$ to denote the information state $I$ corresponding to a history $s$.

A **behavior-form strategy** $\pi(a|I)$ is defined on each information state: $\pi(\cdot|I) \mapsto \Delta_{\mathcal{A}(I)}$ ($\Delta$ is a probability simplex, and $\Delta^\circ$ means the interior of $\Delta$). We further denote the restriction of $\pi$ over $\mathcal{I}_i \subseteq \mathcal{I}$ by $\pi^i$, and thus $\pi = (\pi^i, \pi^{-i})$. If all players follow $\pi$, the **reach probability** of a history $s$ can be computed by $\rho^\pi(s) = \prod_{s'a \sqsubset s} \pi(a|I(s'))$. Thus we have $r^h(I,a) = \sum_{s \in I,a} \rho^\pi(s,a) r^h(s,a) / \sum_{s \in I} \rho^\pi(s)$, where $\rho^\pi(s,a) = \rho^\pi(s)\pi(a|I(s))$. For player 1, the regularized value function is defined as $V_\pi^{h,\tau}(I) = \mathbb{E}\Big[\sum_{h'=h}^H [r^{h'}(I_{h'}, a_{h'}) - \delta(I_{h'}) \cdot \tau \cdot \nabla g(\pi(\cdot|I_{h'}))] | I_h = I\Big]$, and the regularized Q-function as $Q_\pi^{h,\tau}(I,a) = r^h(I,a) + \mathbb{E}_{I'=I(ha),h\in I}[V_\pi^{h+1,\tau}(I')]$, where $\delta(I) = 2 \cdot \mathbf{1}_{1=\mathcal{P}(\mathrm{I})} - 1$, and $g$ is a continuously differentiable regularization function. The un-regularized value functions $V_\pi^h(I), Q_\pi^h(I,a)$ can be obtained by setting $\tau = 0$. The value functions of player 2 are the negative one of player 1. Given a policy $\pi$, we define $\mathcal{E}_\tau(\pi) = \max_p[V_{p^i,\pi^{-i}}^{1,\tau}(s_{\text{init}}) - V_{\pi^i,p^{-i}}^{1,\tau}(s_{\text{init}})]$, and policy $\pi$ is said to be a **regularized equilibrium** (or **Nash equilibrium** if $\tau = 0$) if $\mathcal{E}_\tau(\pi) = 0$, or $\epsilon$-regularized equilibrium if $\mathcal{E}_\tau(\pi) \leq \epsilon$.

## 2.2 NORMAL-FORM GAMES AND FOLLOW-THE-REGULARIZED-LEADER

NFGs are strict sub-class of EFGs, where only one stage exists and all the players act simultaneously. each player $i$ selects action $a^i \in \mathcal{A}$, and then player 1 receives a reward $r(a^1, a^2) \in [0,1]$ while player 2 receives a reward $-r(a^1, a^2)$. For a given policy $\pi = (\pi^1, \pi^2) \in \prod_{i=1}^2 \Delta_{\mathcal{A}}$, the Q-function for player 1 is defined as $Q_\pi(a^1) = \mathbb{E}_{a^2 \sim \pi^2}[r(a^1, a^2)]$, and the value function as $V_\pi = \mathbb{E}_{a^1 \sim \pi^1}[Q_\pi(a^1)]$. The value functions of player 2 are the negative values of player 1.

FTRL [2] is an intuitive algorithm: for player $i$, at each time step it maximizes the sum of the past returns $y_t^i = \int_0^t (2 \cdot \mathbf{1}_{1=i} - 1)Q_{\pi_k} dk$ with a regularization $\psi : \Delta_{\mathcal{A}} \to \mathbb{R}$, i.e., $\pi_{t+1}^i = \arg\max_{p \in \Delta_{\mathcal{A}}}[\eta\langle p, y_t^i\rangle - \psi(p)]$ where $\langle \cdot, \cdot \rangle$ means inner product and $\eta > 0$ is the learning rate.

## 2.3 OTHER NOTATIONS

For a strictly convex and continuously differentiable function $\psi$, we denote the Bregman divergence as $D_\psi(p,q) = \psi(p) - \psi(q) - \langle \nabla\psi(q), p - q \rangle$, and the Kullback-Leibler divergence (i.e., Bregman divergence with $\psi(p) = \sum_a p(a) \ln p(a)$) as $D_{\text{KL}}$. Then, we say that $\psi$ is $\lambda$-strongly convex with respect to $\|\cdot\|$ if $D_\psi(p,q) \geq \frac{\lambda}{2}\|p-q\|^2$, and $g$ is $\lambda$-strongly convex relative to $\psi$ if $\langle \nabla g(p) - \nabla g(q), p - q \rangle \geq \lambda\langle \nabla\psi(p) - \nabla\psi(q), p - q \rangle$. Note that $\psi$ and $D_\psi(\cdot, q)$ is 1-strongly convex relative to $\psi$.

## 3 STABILIZE THE LEARNING DYNAMICS VIA REGULARIZATION

This section utilizes the regularization to stabilize the learning dynamics of FTRL in general games, and presents a comprehensive study of its last-iterate convergence in NFGs and EFGs. We defer all the proofs of our theoretical results to Appendix A.

## 3.1 LAST-ITERATE CONVERGENCE IN NFGS

Since the learning dynamics of FTRL will converge in potential games (Héliou et al., 2017) but cycle in Hamiltonian games (Mertikopoulos et al., 2018; Balduzzi et al., 2018), an intuitive idea to stabilize the FTRL dynamics is to add an extra potential component to the underlying game, i.e., to enhance the potential component of the original game. In this subsection, we present this potential enhancement method in NFGs.

---

[2]See Appendix C for a more detailed presentation of FTRL.

With an arbitrary potential function $g$, we consider the potential-enhancement optimization problem:

$$\max_{\pi^1 \in \Delta_\mathcal{A}} \min_{\pi^2 \in \Delta_\mathcal{A}} V_{\pi^1, \pi^2} - \tau g(\pi^1, \pi^2), \tag{1}$$

where $\tau > 0$ is a weight parameter to control the strength of the additional potential component, and thus the original game can be obtained by setting $\tau = 0$. We further consider a decentralized potential function, i.e., $g(\pi^1, \pi^2) = g^1(\pi^1) - g^2(\pi^2)$, in terms of ease-of-use. It can be found that problem (1) is a generalization of entropy-regularized problem by setting $g^i(p) = \langle p, \ln p \rangle$. Inspired by that, we develop **RegFTRL** by incorporating the additional potential function into FTRL:

$$\pi_t^i = \arg\max_{p \in \Delta_A}[\eta \langle p, y_t^i \rangle - \psi(p)], \tag{2}$$

$$y_t^i(a) = \int_0^t \left[\delta^i Q_{\pi_k}(a) - \tau[\nabla g^i(\pi_k^i)]_a\right]dk, \quad \delta^i = 2 \cdot \mathbf{1}_{1=i} - 1,$$

where $\eta > 0$ is the learning rate, and the regularization function $\psi : \Delta_\mathcal{A} \to \mathbb{R}$ is strictly convex and continuously differentiable on $\Delta_\mathcal{A}$. Note that $\int_0^t \left[\delta^i Q_{\pi_k}(a) - \tau[\nabla g^i(\pi_k^i)]_a\right]dk = \sum_{k=0}^{t-1} \left[\delta^i Q_{\pi_k}(a) - \tau[\nabla g^i(\pi_k^i)]_a\right]$ under discrete-time settings, and FTRL can be obtained by setting $\tau = 0$. It can be found that the learning dynamics of RegFTRL in the original game is equivalent with the FTRL dynamics in the potential-enhancement optimization problem (1).

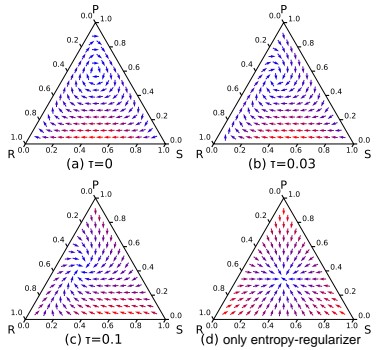

.5In RegFTRL, the extra potential term is expected to force the dynamics to escape from cycles. Figure 2 shows this insight visually, which describes RegFTRL dynamics in a simple Hamiltonian game of biased Rock-Paper-Scissors. Here we take $g^i(\pi^i) = D_{\mathrm{KL}}(\mu^i, \pi^i)$ ($\mu$ is an uniform strategy) as an example. As can be seen in Figure 2 (a), the FTRL dynamics cycle and fail to converge to the interior Nash equilibrium, while the RegFTRL dynamics with only the potential term (i.e., $y_t^i = -\int_0^t \nabla g^i(\pi_k^i)dk$) directly converge to a stationary point in Figure 2 (d), although this stationary point is independent of the original game. However, with some small weight parameters, RegFTRL flows towards a stationary point near the Nash equilibrium as shown in Figure 2 (b) and (c).

Figure 2: The vector fields of RegFTRL with varying weights in biased Rock-Paper-Scissors.

We now analyze the theoretical properties of RegFTRL. Before that, we make some necessary assumptions.

**Assumption 1** (Well Defined). *Assume $\psi$ is 1-strongly convex with respect to $\|\cdot\|$ and $\pi_t \in \mathcal{B} = \mathcal{B}^1 \times \mathcal{B}^2 \subseteq \prod_{i=1}^2 \Delta_\mathcal{A}^\circ$, where $\{\pi_t\}_{t \geq 0}$ is generated by RegFTRL.*

**Assumption 2.** *For $i \in \{1, 2\}$, assume $g^i$ is continuously differentiable and $\lambda$-strongly convex relative to $\psi$ over $\Delta_\mathcal{A}^\circ$. This also implies that $\nabla g^i$ is L-smooth over $\mathcal{B}$, i.e., $\|\nabla g^i(p) - \nabla g^i(q)\| \leq L\|p - q\|$ for $\forall p, q \in \mathcal{B}$. Furthermore, we assume $g^i$ has an interior minimum point $\mu^i \in \mathcal{B}^i$, and re-denote $g^i$ as $g_\mu^i$ for the sake of clarity. We call this minimum point $\mu$ as **reference strategy**.*

**Assumption 3** (Regularized Equilibrium). *Assume $\pi_\mu \in \mathcal{B}$ is the interior stationary point of continuous-time RegFTRL dynamics with $\psi(p) = \langle p, \ln p \rangle$ and $g_\mu$.*

Assumption 1 is to ensure that $\pi_t$ generated by RegFTRL is well defined. This assumption is also required in MMD (Sokota et al., 2023). Assumption 2 allows RegFTRL to get fast convergence via leveraging the curvature of $g_\mu$, and then the potential enhancement method actually is the regularization technique. And Assumption 3 is guaranteed when $g_\mu^i(\pi) = D_{\mathrm{KL}}(\mu^i, \pi^i)$ or $g_\mu^i(\pi) = D_{\mathrm{KL}}(\pi^i, \mu^i)$ (more details refer to Appendix A.1). Under these assumptions, we first present the properties of regularized equilibrium, and then give the linear convergence guarantees of RegFTRL without the uniqueness condition in both continuous-time and discrete-time settings.

**Theorem 1.** *Under Assumption 1~3, regularized equilibrium $\pi_\mu \in \mathcal{B}$ satisfies: (1) $\pi_\mu$ is unique; (2) $\pi_\mu$ is an $\epsilon$-Nash equilibrium, where $\epsilon = \mathcal{E}(\pi_\mu) = \tau \sum_{i=1}^2 \left(\max_a[\nabla g^i(\pi_\mu^i)]_a - \langle \pi_\mu^i, \nabla g^i(\pi_\mu^i) \rangle\right) \geq 0$.*

**Theorem 2.** *Let Assumption 1~3 hold. Then,*
*(**continuous-time**) $\pi_t$ generated by continuous-time version of RegFTRL dynamics satisfies:*

$$D_\psi(\pi_\mu, \pi_t) \leq D_\psi(\pi_\mu, \pi_0) \exp(-\eta\tau\lambda \cdot t).$$

**(discrete-time)** $\pi_t$ generated by discrete-time version of RegFTRL dynamics satisfies:

$$D_\psi(\pi_\mu, \pi_t) \leq D_\psi(\pi_\mu, \pi_0)(1 + \eta\tau\lambda)^{-t}, \text{ and } \mathcal{E}(\pi_t) \leq \mathcal{E}(\pi_\mu) + 2\sqrt{D_\psi(\pi_\mu, \pi_0)}(1 + \eta\tau\lambda)^{-t/2},$$

if $\psi(p) = \langle p, \ln p \rangle$ and $0 < \eta \leq \frac{\tau\lambda}{\tilde{L}^2}$, where $\tilde{L} = \max\{\tau L, 1\}$.

**Remark 1.** *Theorem 1 implies that $\pi_\mu$ is a Nash equilibrium if $\mathcal{E}(\pi_\mu) = 0$, which means $\pi_\mu = \mu$. Additionally, by combining Theorem 1 and Theorem 2, it can be also found that the weight parameter $\tau$ introduces a trade-off between the speed of convergence and the bias in the Nash equilibrium. These observations inspire us to develop the two approaches mentioned in Section 3.3 to reach an exact Nash equilibrium.*

**Remark 2.** *Take the QRE as an example, Theorem 2 implies that discrete-time RegFTRL is guaranteed to find an $\epsilon$-QRE in $\mathcal{O}(\frac{1}{\ln(1+\eta\tau)} \ln \frac{1}{\epsilon})$ iterations. Besides, FTRL-SP (Abe et al., 2023) and entropy-regularized OMWU (Cen et al., 2021) both require $\mathcal{O}(\frac{1}{-\ln(1-\eta\tau/2)} \ln \frac{1}{\epsilon})$ iterations.*

## 3.2 Last-Iterate Convergence in EFGs

This subsection generalizes ReFTRL to EFGs. We first present the RegFTRL algorithm.

**Algorithm 1** (RegFTRL in EFGs). *At any information state $I \in \mathcal{I}$, the policy updated rule is:*

$$\pi_t(\cdot|I) = \arg\max_{p \in \Delta_{\mathcal{A}(I)}}[\eta\langle p, y_t^{h,\tau}(I, \cdot)\rangle - \psi(p)], \tag{3}$$

$$y_t^{h,\tau}(I, a) = \sum_{k=0}^{t-1}\left[\delta(I)Q_k^{h,\tau}(I, a) - \tau[\nabla g_\mu^I(\pi_k)]_a\right],$$

*where $g_\mu^I(\pi_k) := g_\mu(\pi_k(\cdot|I))$, and update the value functions as:*

$$\begin{cases} Q_0 = 0, V_0(I) = -\delta(I)\tau\langle \pi_0(\cdot|I), \nabla g_\mu^I(\pi_0)\rangle \\ Q_t^{h,\tau}(I, a) = r^h(I, a) + \mathbb{E}_{I'=I(ha),h\in I}[V_{t-1}^{h+1,\tau}(I')] \\ V_t^{h,\tau}(I) = (1-\alpha_t)V_{t-1}^{h,\tau}(I) + \alpha_t\sum_a \pi_t(a|I)[Q_t^{h,\tau}(I, a) - \delta(I)\tau \cdot \nabla[g_\mu^I(\pi_t)]_a]. \end{cases} \tag{4}$$

It can be found that Algorithm 1 conforms to the actor-critic framework, wherein the actor is responsible for policy updates through the utilization of RegFTRL (as delineated in Eq. (2)), while the critic undertakes the task of value function updates on a relatively slower timescale. Given the intricate structural complexities introduced by the extensive-form format inherent in EFGs, particularly the non-concave nature of the value functions within the behavior-form representation, we focus on providing the convergence guarantee for a specialized instance of ReFTRL, wherein $g_\mu^I(\pi) = D_{\text{KL}}^I(\pi, \mu)$ and $\psi(\pi) = \langle \pi, \ln \pi \rangle$. It is noteworthy that the entropy-regularized MWU is readily attainable by setting the reference strategy $\mu$ as a uniform strategy.

**Theorem 3.** *Let $g_\mu^I(\pi) = D_{KL}^I(\pi, \mu) := D_{KL}(\pi(\cdot|I), \mu(\cdot|I))$, $\mu \geq \frac{c}{|\mathcal{A}|}$, $c \in (0, 1]$. With $0 < \eta\tau \leq \frac{2}{3}$, $0 < \tau \leq \frac{1}{\max\{1, 2\ln(|\mathcal{A}|/c)\}}$, and $\alpha_t = \eta\tau$, Algorithm 1 satisfies:*

$$\max_{I,a}|Q_{\pi_\mu}^{h,\tau} - Q_t^{h,\tau}| \leq (1 - \eta\tau)^{t-T_h}t^{H-h}, \text{ and } \mathcal{E}_\tau(\pi_t) \leq 8(1 - \eta\tau)^{t-T_1}\left(\frac{5}{3}H + 3t^H\right),$$

*where $t \geq T_h := (H - h)T_s$, $T_s = \lceil \frac{1}{\eta\tau} \ln 2(9H + 19) \rceil$.*

**Remark 3.** *Minimizing the bound over $\eta$, Theorem 3 implies that Algorithm 1 is guaranteed to find an $\epsilon$-QRE in $\tilde{\mathcal{O}}(\frac{H}{\tau} \ln \frac{1}{\epsilon})$ iterations, and thus find an $\epsilon$-Nash equilibrium in $\tilde{\mathcal{O}}(\frac{H^2}{\epsilon})$ iterations by setting $\tau = \mathcal{O}(\frac{\epsilon}{H \ln(|\mathcal{A}|/c)})$ since $\mathcal{E}(\pi) \leq \mathcal{E}_\tau(\pi) + 2\tau H \ln(|\mathcal{A}|/c)$.*

## 3.3 Convergence to an Exact Nash Equilibrium

Section 3.1 and Section 3.2 analyze the theoretical properties of RegFTRL with a fixed reference strategy, and guarantee the linear convergence to an approximated equilibrium. In this subsection, we introduce two approaches to find an exact Nash equilibrium.

The first one is the **annealing approach** that gradually decrease weight parameter $\tau$. This reduction serves the purpose of diminishing the bias associated with the equilibrium. Drawing insights from Theorem 1 and Remark 3, the distance between the regularized equilibrium $\pi_\mu$ and the set of Nash equilibria $\Pi_*$ can be effectively controlled through the manipulation of the weight parameter $\tau$. Nonetheless, it is worth noting that, as implied by Theorem 2 and Theorem 3, the speed of convergence might be adversely affected as the weight parameter $\tau$ undergoes reduction.

Another one is the **adaption approach**, similar to the direct convergence method (Pérolat et al., 2021). As indicated in Remark 1, a Nash equilibrium can be achieved when $\pi_\mu = \mu$, implying that the regularized equilibrium will exhibit closer proximity to $\Pi_*$ if the reference strategy $\mu$ approximates a Nash equilibrium. Therefore, we set reference strategy $\mu$ to $\pi_t$ every $N$ iterations, with the expectation that $\pi_t$ will progressively converge towards a Nash equilibrium. Indeed, with $\mu_k$ denoting the $k$-th reference strategy and a sufficiently large value for $N$, $\pi_t$ converges to $\pi_{\mu_k}$ from Theorem 2 and Theorem 3. Subsequently, the subsequent reference strategy $\mu_{k+1}$ is adjusted to coincide with $\pi_{\mu_k}$. Intuitively, as $k$ increases, $\pi_{\mu_k}$ coincides with $\mu_k$, consequently driving the reference strategies towards convergence with a Nash equilibrium in the underlying game. This intuition is formally substantiated by the following theorem. It is important to highlight that, unlike the annealing approach, the adaption approach obviates the necessity for a diminishing weight parameter $\tau$, thus preserving a consistent convergence rate.

**Theorem 4.** *If $g_\mu(\pi) = D_\phi(\pi, \mu)$ and $g_\mu(\pi) = D_{KL}(\mu, \pi)$, then for any interior point $\mu_0$, the sequence of reference strategies $\{\mu_k\}_{k \geq 0}$ converges to a Nash equilibrium of the original game.*

**Remark 4.** *Moving magnet approach used in MMD (Sokota et al., 2023), wherein the reference strategy is updated as $\mu_{t+1} \propto \mu_t^{1-\beta} \pi_{t+1}^\beta$ ($0 < \beta < 1$), is also an efficient means to promote convergence to a Nash equilibrium. However, the reference strategy updated in this form can be perceived as an average strategy in a certain sense, potentially limiting its practical applicability.*

## 4 FOLLOWMU: A PRACTICAL IMPLEMENTATION OF REGFTRL

This section proposes a novel model-free reinforcement learning algorithm, named FollowMu (**Follow** the reference strategy $\boldsymbol{\mu}$), which combines ReFTRL with function approximation techniques. We employ the actor-critic framework to develop FollowMu due to its scalability (Sutton & Barto, 2018). Let $A(I, a; \theta_t)$ be the actor network parameterized by $\theta_t$, and $V(I; \omega_t)$ be the critic network parameterized by $\omega_t$. At the time step $t$, the critic network $V(I; \omega_t)$ is trained to approximate the value function $V_{\pi_t}(I)$ of the real-time strategy, and the actor network $A(I, a; \theta_t)$ is trained to fit the cumulative advantage function of past iterations plus the Q-function of the current-iterate strategy (with the regularized term):

$$A(I, a; \theta_t) \simeq \sum_{k=0}^{t-1} \left[ Q_{\pi_k}(I, a) - V_{\pi_k}(I) - \tau \log \frac{\pi_k(a|I)}{\mu(a|I)} \right] + Q_{\pi_t}(I, a) - \tau \log \frac{\pi_t(a|I)}{\mu(a|I)}$$

$$\simeq [A(I, a; \theta_{t-1}) - V(I; \omega_{t-1})] + G - \tau \log \frac{\pi_t(a|I)}{\mu(a|I)}, \tag{5}$$

where $G$ is the empirical estimator of $Q_{\pi_t}(I, a)$. Then, if we take $\psi$ to be the entropy regularizer, the next-iterate strategy can be computed by:

$$\pi_{t+1}(a|I) \propto \exp(z_t(I, a)), \quad z_t(I, a) \simeq A(I, a; \theta_t) - V(I; \omega_t). \tag{6}$$

Here we employ the advantage function $Q_{\pi_t}(I, a) - V_{\pi_t}(I)$, as a substitution for the Q-function $Q_{\pi_t}(I, a)$, a choice made for the sake of enhancing numerical stability and robustness. Despite this alteration, the strategy update formulation in the manner of Eq.(6) remains equivalent to the updated strategy employed in RegFTRL, attributed to the shift-invariant nature inherent in the softmax function. Meanwhile, the reference strategy will be updated $\mu \leftarrow \pi_t$ every $N$ iterations. More details of FollowMu are presented in the Appendix B.1.

## 5 EXPERIMENTS

In this section, we validate our methods on NFGs and EFGs utilizing the exploitability metric (i.e., $\mathcal{E}(\pi)$) under two experimental settings, i.e., full-information feedback setting and neural-based sample setting. In full-information feedback setting, we evaluate the performance of RegFTRL as a

Nash equilibrium solver, employing annealing approach and adaption approach. Note that the potential function $g_\mu$ in RegFTRL is set to $g_\mu(\pi) = D_{\mathrm{KL}}(\pi, \mu)$. In addition to this, we also consider moment projection $g_\mu(\pi) = D_{\mathrm{KL}}(\mu, \pi)$ and $L_2$ norm $g_\mu(\pi) = \frac{1}{2}\|\pi - \mu\|_2^2$ for examining the impact brought by different regularization. We abbreviate RegFTRL equipped with $g_\mu(\pi) = D_{\mathrm{KL}}(\mu, \pi)$ as M-RegFTRL, and RegFTRL with $g_\mu(\pi) = \frac{1}{2}\|\pi - \mu\|_2^2$ as 2-RegFTRL. In neural-based sample setting, we assess the efficacy of FollowMu as a deep multi-agent reinforcement learning algorithm through self-play. Further details about the experimental settings are included in Appendix B.2.

## 5.1 Full-Information Feedback Setting

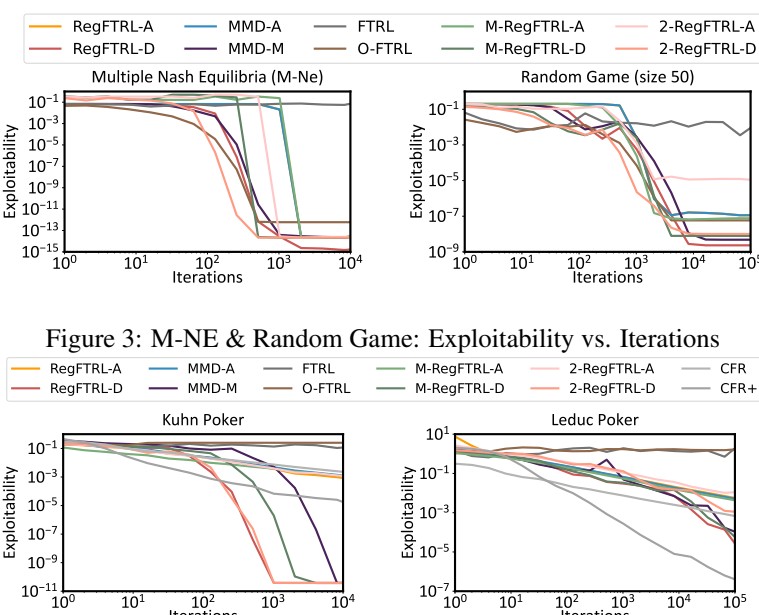

Figure 3: M-NE & Random Game: Exploitability vs. Iterations

Figure 4: Kuhn & Leduc Poker: Exploitability vs. Iterations

In this case, we compare the performances of RegFTRL-A (abbr. RegFTRL with annealing approach), RegFTRL-D (abbr. RegFTRL with adaption approach) with baselines: (i) FTRL, (ii) O-FTRL (abbr. optimistic FTRL), (iii) CFR (Zinkevich et al., 2007), (iv)MMD-A (abbr. MMD with annealing weight), and (v) MMD-M (abbr. MMD with moving magnet). For NFGs, we focus on two games: Multiple Nash Equilibria (abbr. M-Ne) and a random utility game with 50 actions. M-Ne, as introduced in the prior work (Wei et al., 2021), is characterized by a set of Nash equilibria. For the random utility game, the $50 \times 50$ payoff matrix is drawn from a standard Gaussian distribution in an i.i.d. manner. For EFGs, we consider games implemented in OpenSpiel (Lanctot et al., 2019): Kuhn Poker and Leduc Poker, with 54 and 9300 non-terminal histories, respectively.

Figure 3 presents the NFG results. Across all games considered, it is evident that FTRL fails to converge to an equilibrium. Conversely, all other algorithms consistently demonstrate linear convergence rates, aligning with theoretical guarantees. This observation underscores the significant impact of the optimistic update paradigm and regularization techniques in facilitating last-iterate convergence. It is noteworthy that RegFTRL shares mathematical equivalence with MMD within the NFG context. Consequently, RegFTRL-A exhibits performance comparable to that of MMD-A. However, an interesting contrast emerges between RegFTRL-D and MMD-M, with the former displaying superior performance. This discrepancy can potentially be attributed to the reference strategy updated by the moving magnet approach, which retains past-iterate strategies, consequently causing it to deviate from the Nash equilibrium.

Figure 4 provides the results observed within Kuhn & Leduc Poker. Unlike the performances in NFGs, in both Poker games, O-FTRL performs poorly, which might be attributed to the behavior-form based implement. In contrast, despite the absence of theoretical convergence guarantees under EFGs, both M-RegFTRL and 2-RegFTRL exhibit an exponentially fast convergence rate. This outcome underscores their potential utility in EFGs, despite the inherent lack of formal guarantees.

Furthermore, an interesting trend emerges wherein the adaption approach consistently outperforms the annealing approach in both NFGs and EFGs. This phenomenon can be elucidated by referring to Theorem 2 and Theorem 3, which indicate that a decaying weight parameter $\tau$ can bring a slower convergence rate, aligning with our empirical observations.

## 5.2 NEURAL-BASED SAMPLE SETTING

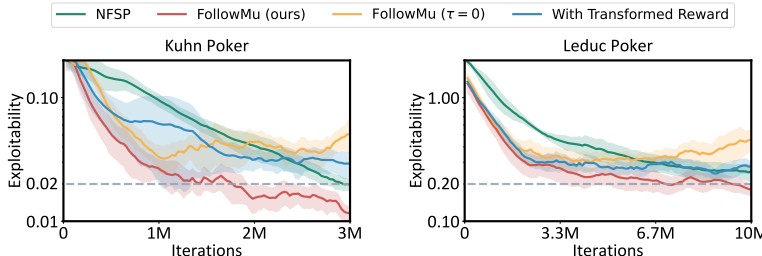

Figure 5: Kuhn & Leduc Poker: Exploitability vs. Iterations

Table 1: Mean$\pm$standard deviation of the approximate exploitability on Phantom Tic-Tac-Toe.

|  | FollowMu | NFSP | PPO | Uniform Agent |
|---|---|---|---|---|
| 1M step | $0.39 \pm 0.02$ | $0.91 \pm 0.04$ | $0.94 \pm 0.03$ | $0.79 \pm 0.02$ |
| 10M step | $0.21 \pm 0.04$ | $0.80 \pm 0.03$ | $0.91 \pm 0.03$ | $0.79 \pm 0.02$ |

In this case, we validate our practical implement of RegFTRL, i.e., FollowMu, can work effectively with function approximator while preserving the theoretical guarantees. Within both Poker benchmarks, as depicted in Figure 5, we conduct a comparative analysis involving FollowMu, NFSP (Heinrich & Silver, 2016), FollowMu without regularization (i.e., practical implement of FTRL), and FollowMu with transformed reward (Pérolat et al., 2021). Our observations reveal that FollowMu consistently outperforms the other baselines in terms of exploitability. The noteworthy disparity between FollowMu and FollowMu ($\tau = 0$) aligns with the learning dynamics associated with ReFTRL and FTRL, substantiating the effectiveness of the incorporated regularization term. Furthermore, our findings indicate that FollowMu surpasses FollowMu with the transformed reward. This distinction may be attributed to the fact that FollowMu introduces the regularization term at the return level, mitigating the cumulative sum effect encountered at the reward level.

Table 1 reports the performances of FollowMu with NFSP, PPO (Schulman et al., 2017), and Uniform Agent (employing a uniform strategy consistently) on Phantom Tic-Tac-Toe. Phantom Tic-Tac-Toe, which is also implemented in OpenSpiel, is an imperfect-information game where the winner receives a payoff of $+1$ and the losing player receives $-1$. The evaluation of approximate exploitability in Phantom Tic-Tac-Toe is computed through a trained DQN best response, owing to the substantial scale of the game. The outcomes underscore that both FollowMu and NFSP exhibit enhanced performance following 10 million steps of training compared to their performance after 1 million steps. In contrast, PPO exhibits negligible improvement, consistent with the fact that it is designed for single-agent environments. Notably, FollowMu emerges as the top performer, significantly outperforming the baseline methods.

## 6 CONCLUSION

In this paper, we introduce RegFTRL, an algorithm that devised to enhance the stability of FTRL dynamics through a general-case regularization, and establish the last-iterate linear convergence in both NFGs and EFGs, without either the uniqueness condition or the optimistic update paradigm. Furthermore, our investigation extends to probing the feasibility of achieving convergence towards an exact Nash equilibrium through two straightforward yet highly efficient approaches. Additionally, we propose a model-free reinforcement learning algorithm for zero-sum games, named FollowMu, which is theoretically justified as it is derived from RegFTRL. The numerical simulation reveals RegFTRL outperforms FTRL and O-FTRL in various zero-sum games, and demonstrates that FollowMu attains favorable performance levels, underscoring its practical utility.

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
