# SUPPLEMENTARY MATERIAL FOR REGFTRL: REGULARIZATION IS ENOUGH FOR LAST-ITERATE CONVERGENCE IN ZERO-SUM GAMES

## CONTENTS

# A  TECHNICAL PROOFS

## A.1  REGFTRL IN NFGS

We list the assumptions again here and begin with an explanation of the rationality of the assumption of the regularized equilibrium.

**Assumption 1** (Well Defined). *Assume $\psi$ is 1-strongly convex with respect to $\|\cdot\|$ and $\pi_t \in \mathcal{B} = \mathcal{B}^1 \times \mathcal{B}^2 \subseteq \prod_{i=1}^2 \Delta_{\mathcal{A}}^\circ$, where $\{\pi_t\}_{t\geq 0}$ is generated by RegFTRL.*

**Assumption 2.** *For $i \in \{1,2\}$, assume $g^i$ is continuously differentiable and $\lambda$-strongly convex relative to $\psi$ over $\Delta_{\mathcal{A}}^\circ$. This also implies that $\nabla g^i$ is L-smooth over $\mathcal{B}$, i.e., $\|\nabla g^i(p) - \nabla g^i(q)\| \leq L\|p - q\|$ for $\forall p, q \in \mathcal{B}$. Furthermore, we assume $g^i$ has an interior minimum point $\mu^i \in \mathcal{B}^i$, and re-denote $g^i$ as $g_\mu^i$ for the sake of clarity. We call this minimum point $\mu$ as **reference strategy**.*

**Assumption 3** (Regularized Equilibrium). *Assume $\pi_\mu \in \mathcal{B}$ is the interior stationary point of continuous-time RegFTRL dynamics with $\psi(p) = \langle p, \ln p \rangle$ and $g_\mu$.*

**Proposition 1.** *Assumption 3 is guaranteed when $g_\mu(\pi) = D_{KL}(\pi, \mu)$ and $g_\mu(\pi) = D_{KL}(\mu, \pi)$.*

*Proof.* For any interior reference strategy $\mu$, if there exists an action $a_0$ such that $\pi_\mu(a_0) = 0$, then $Q_{\pi_\mu}(a_0) - \tau[\nabla g_\mu(\pi_\mu)]_{a_0} < Q_{\pi_\mu}(a_*) - \tau[\nabla g_\mu(\pi_\mu)]_{a_*}$ holds for any $a_* \in \{a \in \mathcal{A} | \pi_\mu(a) > 0\}$. However, for $g_\mu(\pi) = D_{KL}(\pi, \mu)$ and $g_\mu(\pi) = D_{KL}(\mu, \pi)$, we have:

$$Q_{\pi_\mu}(a_0) - \tau[\nabla g_\mu(\pi_\mu)]_{a_0}$$
$$= \begin{cases} Q_{\pi_\mu}(a_0) - \tau(\ln \frac{\pi_\mu(a_0)}{\mu(a_0)} + 1) = \infty, & \text{if } g_\mu(\pi) = D_{KL}(\pi, \mu), \\ Q_{\pi_\mu}(a_0) + \tau \frac{\mu(a_0)}{\pi_\mu(a_0)} = \infty, & \text{if } g_\mu(\pi) = D_{KL}(\mu, \pi), \end{cases}$$

which is a contradiction, and thus Assumption 3 holds. $\qquad\square$

We continue by proving the properties of regularized equilibrium, and the last-iterate convergence of continuous-time version of RegFTRL in NFGs. For convenience, we denote $Q_\pi^1(a) = Q_\pi(a)$ and $Q_\pi^2(a) = -Q_\pi(a)$ in the context of NFGs.

**Theorem 1.** *Under Assumption 1~3, regularized equilibrium $\pi_\mu \in \mathcal{B}$ satisfies: (1) $\pi_\mu$ is unique; (2) $\pi_\mu$ is an $\epsilon$-Nash equilibrium, where $\epsilon = \mathcal{E}(\pi_\mu) = \tau \sum_{i=1}^2 \left( \max_a[\nabla g^i(\pi_\mu^i)]_a - \langle \pi_\mu^i, \nabla g^i(\pi_\mu^i) \rangle \right) \geq 0$.*

*Proof.* Note that Theorem 2 holds for all regularized equilibria. This means that $\pi_\mu$ is unique if the weight parameter $\tau$ and the reference strategy $\mu$ are fixed. Thus we only need to provide the proofs of the second statement that $\pi_\mu$ is an $\epsilon$-Nash equilibrium.

Since regularized equilibrium is the interior stationary point of continuous-time RegFTRL dynamics with $\psi(p) = \langle p, \ln p \rangle$. By the method of Lagrange multiplier, it can be found that the dynamics defined by continuous-time RegFTRL with the entropy regularizer is equivalent to the following dynamics:

$$\frac{d}{dt}\pi_t^i(a) = \pi_t^i(a)\left(Q_{\pi_t}^i(a) - \tau[\nabla g_\mu^i(\pi_t^i)]_a - V_{\pi_t}^i + \tau\langle \pi_t^i, \nabla g_\mu^i(\pi_t^i) \rangle\right) \tag{1}$$

From Assumption 3, we have:

$$Q_{\pi_\mu}^i(a) - \tau[\nabla g_\mu^i(\pi_\mu^i)]_a - V_{\pi_\mu}^i + \tau\langle \pi_\mu^i, \nabla g_\mu^i(\pi_\mu^i) \rangle = 0 \tag{2}$$

Therefore, we have:

$$\mathcal{E}(\pi_\mu) = \sum_{i=1}^2 \max_{p^i \in \Delta_{\mathcal{A}}} V_{p,\pi_\mu^{-i}}^i = \sum_{i=1}^2 \left[ \max_{p^i \in \Delta_{\mathcal{A}}} V_{p,\pi_\mu^{-i}}^i - V_{\pi_\mu}^i \right]$$

$$= \sum_{i=1}^2 \left[ \max_{a \in \mathcal{A}} Q_{\pi_\mu}^i(a) - V_{\pi_\mu}^i \right] \leq \tau \sum_{i=1}^2 \left[ \max_{a \in \mathcal{A}}[\nabla g_\mu^i(\pi_\mu^i)]_a - \langle \pi_\mu^i, \nabla g_\mu^i(\pi_\mu^i) \rangle \right]$$

Thus, the proof is completed. $\qquad\square$

**Lemma 1.** $\mathcal{E}$ *can be bounded as follows:*

$$\mathcal{E}(\pi_t) \le \mathcal{E}(\pi_\mu) + \sqrt{D_{KL}(\pi_\mu, \pi_t)}.$$

*Proof.* From the definition of $\mathcal{E}$, we have:

$$
\begin{aligned}
\mathcal{E}(\pi_t) &= \sum_{i=1}^{2} \max_{p^i \in \Delta_\mathcal{A}} V^i_{p, \pi_t^{-i}} \\
&= \sum_{i=1}^{2} \Big( \max_{p^i \in \Delta_\mathcal{A}} V^i_{p, \pi_\mu^{-i}} + \max_{p^i \in \Delta_\mathcal{A}} V^i_{p, \pi_t^{-i}} - \max_{p^i \in \Delta_\mathcal{A}} V^i_{p, \pi_\mu^{-i}} \Big) \\
&= \mathcal{E}(\pi_\mu) + \sum_{i=1}^{2} \Big( \max_{p^i \in \Delta_\mathcal{A}} V^i_{p, \pi_t^{-i}} - \max_{p^i \in \Delta_\mathcal{A}} V^i_{p, \pi_\mu^{-i}} \Big) \\
&\le \mathcal{E}(\pi_\mu) + \sum_{i=1}^{2} \max_{p^i \in \Delta_\mathcal{A}} \big( V^i_{p, \pi_t^{-i}} - V^i_{p, \pi_\mu^{-i}} \big) \\
&\overset{\text{Hölder's inequality}}{\le} \mathcal{E}(\pi_\mu) + \sum_{i=1}^{2} \big( \|\pi_\mu^i - \pi_t^i\|_1 \max_{p^{-i} \in \Delta_\mathcal{A}} \|Q^i_{\pi_t, p^{-i}}\|_\infty \big) \\
&\overset{\text{Pinsker inequality}}{\le} \mathcal{E}(\pi_\mu) + \sum_{i=1}^{2} \sqrt{2 D_{\text{KL}}(\pi_\mu^i, \pi_t^i)} \\
&\overset{\text{Cauchy inequality}}{\le} \mathcal{E}(\pi_\mu) + \sqrt{2} \sqrt{2 \sum_{i=1}^{2} D_{\text{KL}}(\pi_\mu^i, \pi_t^i)} \\
&= \mathcal{E}(\pi_\mu) + 2\sqrt{D_{\text{KL}}(\pi_\mu, \pi_t)}.
\end{aligned}
$$

$\square$

**Theorem 2.** *Let Assumption 1$\sim$3 hold. Then,*
*(**continuous-time**) $\pi_t$ generated by continuous-time version of RegFTRL dynamics satisfies:*

$$D_\psi(\pi_\mu, \pi_t) \le D_\psi(\pi_\mu, \pi_0) \exp(-\eta \tau \lambda \cdot t).$$

*(**discrete-time**) $\pi_t$ generated by discrete-time version of RegFTRL dynamics satisfies:*

$$D_\psi(\pi_\mu, \pi_t) \le D_\psi(\pi_\mu, \pi_0)(1 + \eta\tau\lambda)^{-t}, \text{ and } \mathcal{E}(\pi_t) \le \mathcal{E}(\pi_\mu) + 2\sqrt{D_\psi(\pi_\mu, \pi_0)}(1 + \eta\tau\lambda)^{-t/2},$$

*if $\psi(p) = \langle p, \ln p \rangle$ and $0 < \eta \le \frac{\tau\lambda}{\tilde{L}^2}$, where $\tilde{L} = \max\{\tau L, 1\}$.*

*Proof.* Theorem 2 is the summary of lemma 1, Theorem 3 and Theorem 4, and thus we omit the proofs here. $\square$

### A.1.1 CONTINUOUS-TIME REGFTRL

**Theorem 3.** *Let Assumption 1$\sim$3 hold. Then, $\pi_t$ generated by continuous-time version of RegFTRL dynamics satisfies:*

$$D_\psi(\pi_\mu, \pi_t) \le D_\psi(\pi_\mu, \pi_0) \cdot \exp(-\eta \tau \lambda \cdot t).$$

*Proof.* By lemma C.1 in Abe et al. (2022b), we have:

$$
\begin{aligned}
\frac{d}{dt} D_\psi(\pi_\mu, \pi_t) &= \sum_i \left\langle \frac{d}{dt} y_t^i, \pi_t^i - \pi_\mu^i \right\rangle \\
&= \eta \sum_i \left\langle Q_{\pi_t}^i - \tau \nabla g_\mu^i(\pi_t^i), \pi_t^i - \pi_\mu^i \right\rangle \\
&= \eta \sum_i \left\{ V_{\pi_t}^i - V_{\pi_\mu^i, \pi_t^{-i}}^i - \tau \langle \nabla g_\mu^i(\pi_t^i), \pi_t^i - \pi_\mu^i \rangle \right\} \\
&= \eta \sum_i \left\{ V_{\pi_t^i, \pi_\mu^{-i}}^i - \tau \langle \nabla g_\mu^i(\pi_t^i), \pi_t^i - \pi_\mu^i \rangle \right\} \\
&= \eta \sum_i \left\{ \langle \pi_t^i, Q_{\pi_\mu}^i \rangle - \tau \langle \nabla g_\mu^i(\pi_t^i), \pi_t^i - \pi_\mu^i \rangle \right\} \\
&= \eta \sum_i \left\{ V_{\pi_\mu}^i - \tau \langle \nabla g_\mu^i(\pi_\mu^i), \pi_\mu^i \rangle + \tau \langle \nabla g_\mu^i(\pi_\mu^i), \pi_t^i \rangle - \tau \langle \nabla g_\mu^i(\pi_t^i), \pi_t^i - \pi_\mu^i \rangle \right\} \\
&= -\eta\tau \sum_i \left\{ \langle \nabla g_\mu^i(\pi_\mu^i) - \nabla g_\mu^i(\pi_t^i), \pi_\mu^i - \pi_t^i \rangle \right\} \\
&\leq -\eta\tau\lambda \sum_i \left\{ \langle \nabla \psi(\pi_\mu^i) - \nabla \psi(\pi_t^i), \pi_\mu^i - \pi_t^i \rangle \right\} \\
&= -\eta\tau\lambda \Big[ D_\psi(\pi_\mu, \pi_t) + D_\psi(\pi_t, \pi_\mu) \Big] \\
&\leq -\eta\tau\lambda D_\psi(\pi_\mu, \pi_t).
\end{aligned}
\tag{3}
$$

The sixth equality follows from Eq.(2). Therefore, we have:

$$
D_\psi(\pi_\mu, \pi_t) \leq D_\psi(\pi_\mu, \pi_0) \cdot \exp(-\eta\tau\lambda \cdot t).
$$

Thus, the proof is completed. $\qquad\square$

### A.1.2 DISCRETE-TIME REGFTRL

**Theorem 4.** *Let Assumption 1∼3 hold. Then, $\pi_t$ generated by discrete-time version of RegFTRL dynamics satisfies:*

$$
D_\psi(\pi_\mu, \pi_t) \leq D_\psi(\pi_\mu, \pi_0) \cdot (1 + \eta\tau\lambda)^{-t},
$$

*if $\psi(p) = \langle p, \ln p \rangle$ and $0 < \eta \leq \frac{\tau\lambda}{\tilde{L}^2}$, where $\tilde{L} = \max\{\tau L, 1\}$.*

*Proof.* Let us define $f_\pi^i := Q_\pi^i - \tau \nabla g_\mu^i(\pi^i)$. With $\psi(p) = \langle p, \ln p \rangle$ and the method of Lagrange multiplier, $\pi_t$ generated by RegFTRL satisfies:

$$
\begin{aligned}
& \pi_{t+1}^i(a) \propto \pi_t^i(a) \exp\left\{ \eta \big[ Q_{\pi_t}^i(a) - \tau [\nabla g_\mu^i(\pi_t^i)]_a \big] \right\} \\
\Longleftrightarrow\quad & \pi_{t+1}^i = \arg\max_{p \in \Delta_\mathcal{A}} \left\{ \eta \langle p, f_{\pi_t}^i \rangle - D_\psi(p, \pi_t^i) \right\} \\
\Longleftrightarrow\quad & \langle \eta f_{\pi_t}^i - \nabla \psi(\pi_{t+1}^i) + \nabla \psi(\pi_t^i), \pi^i - \pi_{t+1}^i \rangle \leq 0, \quad \forall \pi^i \in \Delta_\mathcal{A} \\
\Longleftrightarrow\quad & \langle \eta f_{\pi_t}^i, \pi^i - \pi_{t+1}^i \rangle \leq \langle \nabla \psi(\pi_{t+1}^i) - \nabla \psi(\pi_t^i), \pi^i - \pi_{t+1}^i \rangle, \quad \forall \pi^i \in \Delta_\mathcal{A} \\
\Longleftrightarrow\quad & \langle \eta f_{\pi_t}^i, \pi^i - \pi_{t+1}^i \rangle \leq D_{\mathrm{KL}}(\pi^i, \pi_t^i) - D_{\mathrm{KL}}(\pi^i, \pi_{t+1}^i) - D_{\mathrm{KL}}(\pi_{t+1}^i, \pi_t^i), \quad \forall \pi^i \in \Delta_\mathcal{A}
\end{aligned}
$$

The third "$\Longleftrightarrow$" follows from the equivalent first order optimality conditions. Therefore, we have:

$$
D_{\mathrm{KL}}(\pi_\mu^i, \pi_{t+1}^i) \leq D_{\mathrm{KL}}(\pi_\mu^i, \pi_t^i) - D_{\mathrm{KL}}(\pi_{t+1}^i, \pi_t^i) - \eta \langle f_{\pi_t}^i, \pi_\mu^i - \pi_{t+1}^i \rangle.
\tag{4}
$$

On the other hand, we have:

$$
\begin{aligned}
\eta \sum_{i=1}^{2} \left\langle f_{\pi_t}^i, \pi_\mu^i - \pi_t^i \right\rangle =& \eta \sum_{i=1}^{2} \left\langle f_{\pi_\mu}^i, \pi_\mu^i - \pi_t^i \right\rangle + \eta \sum_{i=1}^{2} \left\langle Q_{\pi_t}^i - Q_{\pi_\mu}^i, \pi_\mu^i - \pi_t^i \right\rangle \\
& + \eta\tau \sum_{i=1}^{2} \left\langle \nabla g_\mu^i(\pi_\mu^i) - \nabla g_\mu^i(\pi_t^i), \pi_\mu^i - \pi_t^i \right\rangle \\
\geq& \eta\tau\lambda \sum_{i=1}^{2} \left\langle \nabla \psi(\pi_\mu^i) - \nabla \psi(\pi_t^i), \pi_\mu^i - \pi_t^i \right\rangle \\
=& \eta\tau\lambda \big[ D_{\mathrm{KL}}(\pi_\mu, \pi_t) + D_{\mathrm{KL}}(\pi_t, \pi_\mu) \big].
\end{aligned}
\tag{5}
$$

The inequality follows from the fact that

$$
\left\langle f_{\pi_\mu}^i, \pi_\mu^i - \pi_t^i \right\rangle = \left\langle V_{\pi_\mu}^i \mathbf{1} - \tau\langle \pi_\mu^i, \nabla g_\mu^i(\pi_\mu^i) \rangle \mathbf{1}, \pi_\mu^i - \pi_t^i \right\rangle = 0,
$$

and

$$
\sum_{i=1}^{2} \left\langle Q_{\pi_t}^i - Q_{\pi_\mu}^i, \pi_\mu^i - \pi_t^i \right\rangle = \sum_{i=1}^{2} (V_{\pi_\mu^i, \pi_t^{-i}}^i - V_{\pi_\mu}^i - V_{\pi_t}^i + V_{\pi_t^i, \pi_\mu^{-i}}^i) = 0.
$$

By combining Eq.(4) and Eq.(5), we have:

$$
\begin{aligned}
D_{\mathrm{KL}}(\pi_\mu, \pi_{t+1}) \leq& D_{\mathrm{KL}}(\pi_\mu, \pi_t) - D_{\mathrm{KL}}(\pi_{t+1}, \pi_t) - \eta \sum_{i=1}^{2} \left\langle f_{\pi_t}^i, \pi_\mu^i - \pi_{t+1}^i \right\rangle \\
\leq& D_{\mathrm{KL}}(\pi_\mu, \pi_t) - D_{\mathrm{KL}}(\pi_{t+1}, \pi_t) - \eta\tau\lambda D_{\mathrm{KL}}(\pi_\mu, \pi_{t+1}) - \eta\tau\lambda D_{\mathrm{KL}}(\pi_{t+1}, \pi_\mu) \\
& + \eta \sum_{i=1}^{2} \left\langle f_{\pi_{t+1}}^i - f_{\pi_t}^i, \pi_\mu^i - \pi_{t+1}^i \right\rangle \\
\leq& D_{\mathrm{KL}}(\pi_\mu, \pi_t) - D_{\mathrm{KL}}(\pi_{t+1}, \pi_t) - \eta\tau\lambda D_{\mathrm{KL}}(\pi_\mu, \pi_{t+1}) - \eta\tau\lambda D_{\mathrm{KL}}(\pi_{t+1}, \pi_\mu) \\
& + \eta \max\{1, \tau L\} \| \pi_{t+1} - \pi_t \|_1 \cdot \| \pi_{t+1} - \pi_\mu \|_1 \\
\overset{\tilde{L}=\max\{1,\tau L\},\, 2ab \leq \rho a^2 + \frac{b^2}{\rho}}{\leq}& D_{\mathrm{KL}}(\pi_\mu, \pi_t) - D_{\mathrm{KL}}(\pi_{t+1}, \pi_t) - \eta\tau\lambda D_{\mathrm{KL}}(\pi_\mu, \pi_{t+1}) - \eta\tau\lambda D_{\mathrm{KL}}(\pi_{t+1}, \pi_\mu) \\
& + \frac{1}{2} \| \pi_{t+1} - \pi_t \|_1^2 + \frac{\eta^2 \tilde{L}^2}{2} \| \pi_{t+1} - \pi_\mu \|_1^2 \\
\overset{\text{Pinsker inequality}}{\leq}& D_{\mathrm{KL}}(\pi_\mu, \pi_t) - D_{\mathrm{KL}}(\pi_{t+1}, \pi_t) - \eta\tau\lambda D_{\mathrm{KL}}(\pi_\mu, \pi_{t+1}) - \eta\tau\lambda D_{\mathrm{KL}}(\pi_{t+1}, \pi_\mu) \\
& + D_{\mathrm{KL}}(\pi_{t+1}, \pi_t) + \eta^2 \tilde{L}^2 D_{\mathrm{KL}}(\pi_{t+1}, \pi_\mu) \\
\overset{\eta \leq \frac{\tau\lambda}{\tilde{L}^2}}{\leq}& D_{\mathrm{KL}}(\pi_\mu, \pi_t) - \eta\tau\lambda D_{\mathrm{KL}}(\pi_\mu, \pi_{t+1}).
\end{aligned}
\tag{6}
$$

The third inequality follows from the fact that $r(a^1, a^2) \in [0, 1]$ and $\nabla g$ is $L$-smooth. Therefore, we have:

$$
D_{\mathrm{KL}}(\pi_\mu, \pi_{t+1}) \leq \frac{1}{1 + \eta\tau\lambda} D_{\mathrm{KL}}(\pi_\mu, \pi_t).
\tag{7}
$$

Thus, the proof is completed. $\qquad\square$

### A.1.3 CONVERGENCE TO AN EXACT NASH EQUILIBRIUM

We begin with following useful lemmas.

**Lemma 2.** *If $\pi_\mu \neq \mu$, we have $D_\phi(\pi_*, \pi_\mu) < D_\phi(\pi_*, \mu), \forall \pi_* \in \Pi_*$.*

*Proof.* From the definition of the regularized equilibrium, we have

$$D_\phi(\pi_*, \pi_\mu) - D_\phi(\pi_*, \mu) = - D_\phi(\pi_\mu, \mu) - \sum_{i=1}^{2} \langle \nabla\phi(\pi_\mu^i) - \nabla\phi(\mu^i), \pi_*^i - \pi_\mu^i \rangle$$

$$= - D_\phi(\pi_\mu, \mu) - \frac{1}{\tau} \sum_{i=1}^{2} \langle Q_{\pi_\mu}^i, \pi_*^i - \pi_\mu^i \rangle$$

$$\leq - D_\phi(\pi_\mu, \mu) < 0.$$

The inequality follows from the fact that

$$-\sum_{i=1}^{2} \langle Q_{\pi_\mu}^i, \pi_*^i - \pi_\mu^i \rangle = \sum_{i=1}^{2} [V_{\pi_\mu^i, \pi_*^{-i}}^i - V_{\pi_*}^i] \leq 0.$$

Thus, the proof is completed. $\square$

**Lemma 3.** *If $\pi_\mu = \mu$, then $\mu$ is a Nash equilibrium of the original game.*

*Proof.* By the definition of the regularized equilibrium, when $\pi_\mu = \mu$, we have:

$$\pi_\mu^i(a) [Q_{\pi_\mu}^i(a) - \tau[\nabla g_\mu^i(\pi_\mu^i)]_a - V_{\pi_\mu}^i + \tau\langle \pi_\mu^i, \nabla g_\mu^i(\pi_\mu^i) \rangle] = 0$$

$$\overset{\pi_\mu(a)>0}{\Longrightarrow} Q_{\pi_\mu}^i(a) - \tau[\nabla g_\mu^i(\pi_\mu^i)]_a - V_{\pi_\mu}^i + \tau\langle \pi_\mu^i, \nabla g_\mu^i(\pi_\mu^i) \rangle = 0$$

$$\overset{\pi_\mu=\mu}{\Longrightarrow} Q_{\pi_\mu}^i(a) - V_{\pi_\mu}^i = 0$$

Therefore, $V_{\pi_\mu}^i = \max_{a\Delta_\mathcal{A}} Q_{\pi_\mu}^i(a)$ for $i = 1, 2$, which means that each player's strategy is a best response to the strategy of the other player. Thus, $\mu$ is a Nash equilibrium of the original game. $\square$

**Lemma 4.** *For any $k \geq 0$, if $\mu_k \in \prod_{i=1}^{2} \Delta_\mathcal{A}^\circ \backslash \Pi_*$, then $\min_{\pi_*\in\Pi_*} D_\phi(\pi_*, \mu_{k+1}) < \min_{\pi_*\in\Pi_*} D_\phi(\pi_*, \mu_k)$. Otherwise, if $\mu_k \in \Pi_*$, then $\mu_{k+1} = \mu_k \in \Pi_*$.*

*Proof.* From lemma 3, if $\mu \in \prod_{i=1}^{2} \Delta_\mathcal{A}^\circ \backslash \Pi_*$, then we have $\pi_\mu \neq \mu$. Denote $\pi_* = \arg\min_{\pi_*\in\Pi_*} D_\phi(\pi_*, \mu)$. Then from lemma 2, if $\mu \neq \pi_\mu$, we have:

$$\min_{\pi_*\in\Pi_*} D_\phi(\pi_*, \mu) = D_\phi(\pi_*, \mu) > D_\phi(\pi_*, \pi_\mu) \geq \min_{\pi_*\in\Pi_*} D_\phi(\pi_*, \pi_\mu).$$

Therefore, we prove the first statement of the lemma. Then we assume that $\mu \in \Pi_*$ implies $\pi_\mu \neq \mu$. From lemma 2, we have $D_\phi(\pi_*, \mu) > D_\phi(\pi_*, \pi_\mu)$ for any $\pi_* \in \Pi_*$, and thus $0 > D_\phi(\mu, \pi_\mu)$ due to $r \in \Pi_*$. It is a contradiction.

Thus, the proof is completed. $\square$

**Lemma 5.** *Let $F(\mu) = \pi_\mu$ be a map that maps the reference strategy $\mu$ to its corresponding regularized equilibrium $\pi_\mu$. Then, $F$ is continuous.*

*Proof.* For any given reference strategies $\mu, \hat{\mu} \in \prod_i \Delta_\mathcal{A}^\circ$, we denote their associated stationary points as $\pi_\mu, \pi_{\hat{\mu}}$ respectively. Suppose that $\pi_t$ is the updated strategy of continues-time FTRL dynamics with reference strategy $\mu$ and $\psi(p) = \langle p, \ln p \rangle$, then

$$\frac{d}{dt} \pi_t^i(a) = \pi_t^i(a) [Q_{\pi_t}^i(a) - \tau[\nabla g_\mu^i(\pi_t^i)]_a - V_{\pi_t}^i + \tau\langle \pi_t^i, \nabla g_\mu^i(\pi_t^i) \rangle].$$

Therefore, we have

$$\frac{d}{dt} D_{\mathrm{KL}}(\pi_{\hat{\mu}}, \pi_t) = -\sum_{i=1}^{2} \langle \pi_{\hat{\mu}}^i, \frac{1}{\pi_t^i} \frac{d}{dt} \pi_t^i \rangle$$

$$= \sum_{i=1}^{2} \langle \pi_t^i - \pi_{\hat{\mu}}^i, Q_{\pi_t}^i - \tau \nabla g_\mu^i(\pi_t^i) \rangle$$

$$= \underbrace{\sum_{i=1}^{2} \langle \pi_t^i - \pi_{\hat{\mu}}^i, Q_{\pi_t}^i - \tau[\nabla\phi(\pi_t^i) - \nabla\phi(\hat{\mu}^i)] \rangle}_{(1)}$$

$$+ \underbrace{\tau \sum_{i=1}^{2} \langle \pi_t^i - \pi_{\hat{\mu}}^i, \nabla\phi(\mu^i) - \nabla\phi(\hat{\mu}^i) \rangle}_{(2)}.$$

Then,

$$(1) = \sum_{i=1}^{2} V_{\pi_t^i, \pi_{\hat{\mu}}^{-i}} - \tau \sum_{i=1}^{2} \langle \pi_t^i - \pi_{\hat{\mu}}^i, \nabla\phi(\pi_t^i) - \nabla\phi(\hat{\mu}^i) \rangle$$

$$= \sum_{i=1}^{2} \langle \pi_t^i, Q_{\pi_{\hat{\mu}}}^i \rangle - \tau \sum_{i=1}^{2} \langle \pi_t^i - \pi_{\hat{\mu}}^i, \nabla\phi(\pi_t^i) - \nabla\phi(\hat{\mu}^i) \rangle$$

$$= \sum_{i=1}^{2} \langle \pi_t^i, V_{\pi_{\hat{\mu}}}^i \rangle + \tau \sum_{i=1}^{2} \langle \pi_t^i - \pi_{\hat{\mu}}^i, \nabla\phi(\pi_{\hat{\mu}}^i) - \nabla\phi(\hat{\mu}^i) \rangle - \tau \sum_{i=1}^{2} \langle \pi_t^i - \pi_{\hat{\mu}}^i, \nabla\phi(\pi_t^i) - \nabla\phi(\hat{\mu}^i) \rangle$$

$$= \tau \sum_{i=1}^{2} \langle \pi_t^i - \pi_{\hat{\mu}}^i, \nabla\phi(\pi_{\hat{\mu}}^i) - \nabla\phi(\pi_t^i) \rangle$$

$$= -\tau D_\phi(\pi_t, \pi_{\hat{\mu}}) - \tau D_\phi(\pi_{\hat{\mu}}, \pi_t).$$

On the other hand, we have $(2) \leq 2\tau L \|\hat{\mu} - \mu\|$. By setting $\pi_t = \pi_\mu$, we have $\pi_t = \pi_\mu$, for any $t \geq 0$, and thus we have $D_\phi(\pi_\mu, \pi_{\hat{\mu}}) \leq 2L \|\hat{\mu} - \mu\|$, which means that $F$ is continuous.

Thus, the proof is completed. $\qquad\square$

**Theorem 5.** *If $g_\mu(\pi) = D_\phi(\pi, \mu)$ and $g_\mu(\pi) = D_{KL}(\mu, \pi)$, then for any interior point $\mu_0$, the sequence of reference strategies $\{\mu_k\}_{k \geq 0}$ converges to a Nash equilibrium of the original game.*

*Proof.* In the case that $g_\mu(\pi) = D_{\mathrm{KL}}(\mu, \pi)$, RegFTRL is equivalent with M2WU, and thus the convergence result can be guaranteed by Theorem 6.1 in (Abe et al., 2022a). We next provide the proof of the case that $g_\mu(\pi) = D_\phi(\pi, \mu)$. Denote $b = \lim_{k\to\infty} \min_{\pi_* \in \Pi_*} D_\phi(\pi_*, \mu_k) \geq 0$. We next prove that $b = 0$ and thus $\mu_k$ converges to $\Pi_*$.

By contradiction, we suppose that $b > 0$ and define $B = \min_{\pi_* \in \Pi_*} D_\phi(\pi_*, \mu_0)$. From lemma 4, $\min_{\pi_* \in \Pi_*} D_\phi(\pi_*, \mu_k)$ monotonically decreases, and thus each $\mu_k$ falls into the set $\Omega_{b,B} = \{\mu \in \prod_{i=1}^{2} \Delta_{\mathcal{A}}^\circ : b \leq \min_{\pi_* \in \Pi_*} D_\phi(\pi_*, \mu) \leq B\}$. From lemma 5, $\min_{\pi_* \in \Pi_*} D_\phi(\pi_*, \mu)$ is continuous on $\prod_{i=1}^{2} \Delta_{\mathcal{A}(I)}^\circ$, and thus $\Omega_{b,B}$ is a compact set due to the boundedness of $\prod_{i=1}^{2} \Delta_{\mathcal{A}}^\circ$.

From lemma 5, $\Delta V(\mu) := \min_{\pi_* \in \Pi_*} D_\phi(\pi_*, F(\mu)) - \min_{\pi_* \in \Pi_*} D_\phi(\pi_*, \mu)$ is also continuous. Thus $\Delta V(\mu)$ has a maximum over a compact set, i.e., $M = \max_{\mu \in \Omega_{b,B}} \Delta V(\mu)$ exists. From Lemma 4, $M < 0$, and thus we have:

$$\min_{\pi_* \in \Pi_*} D_\phi(\pi_*, \mu_k) = \min_{\pi_* \in \Pi_*} D_\phi(\pi_*, \mu_0) + \sum_{l=0}^{k-1} \Big( \min_{\pi_* \in \Pi_*} D_\phi(\pi_*, \mu_{l+1}) - \min_{\pi_* \in \Pi_*} D_\phi(\pi_*, \mu_l) \Big)$$

$$\leq B + kM.$$

This implies that $\min_{\pi_* \in \Pi_*} D_\phi(\pi_*, \mu_k) < 0$ for $k > \frac{-B}{M}$, which is a contradiction since $\min_{\pi_* \in \Pi_*} D_\phi(\pi_*, \mu_k) \geq 0$.

Thus, the proof is completed. $\qquad\square$

### A.2 REGFTRL IN EFGS

We start with following lemmas, with the notations in Table 1:

Table 1: Notations in EFGs

| NOTATION | DESCRIPTION |
| --- | --- |
| $A$ | the cardinality of action space $|\mathcal{A}|$ |
| $c$ | $c := A \times \min_{a,I} \mu(a|I) \in (0, 1]$ |
| $\tau$ | weight of regularization $0 < \tau \leq \frac{1}{\max\{1, 2\ln(A/c)\}}$ |
| $\|\cdot\|_p$ | $L_p$-norm: $\|x\|_p = (\sum_i |x_i|^p)^{1/p}$ for $p \geq 1$ and $\|x\|_\infty = \max_i |x_i|$ |

**Lemma 6.** *With $0 < \tau \leq \frac{1}{\max\{1, 2\ln(A/c)\}}$, for any $t \geq 0$ and $h \in [H]$, we have:*

$$
\begin{cases}
(h - H)\tau \ln \frac{A}{c} \leq Q_t^{h,\tau}(I, a) \leq (H - h + 1) + \mathbf{1}_{h \neq H} \cdot (H - h - 1)\tau \ln \frac{A}{c} \\
(h - H - 1)\tau \ln \frac{A}{c} \leq V_t^{h,\tau}(I) \leq (H - h + 1) + (H - h)\tau \ln \frac{A}{c}.
\end{cases}
$$

*Proof.* We prove this lemma by induction. For $t = 0$ and any $h \in [H]$, the statement holds trivially by the definition of $Q_0, V_0$. When the statement holds for some $t$, we have

$$
Q_{t+1}^{h,\tau}(I, a) \leq 1 + \max_{I'} V_t^{h+1,\tau}(I') \leq 1 + (H - h) + (H - h - 1)\tau \ln \frac{A}{c}
$$

$$
\leq (H - h + 1) + (H - h - 1)\tau \ln \frac{A}{c}, \quad h = 1, \cdots, H - 1,
$$

$$
Q_{t+1}^{h,\tau}(I, a) \leq 1, \quad h = H.
$$

$$
Q_{t+1}^{h,\tau}(I, a) \geq 0 + \min_{I'} V_t^{h+1,\tau}(I') \geq (h - H)\tau \ln \frac{A}{c}.
$$

This complete the proof of $Q_t^{h,\tau}$. The proof of $V_t^{h,\tau}$ can be shown with a similar proof and is therefore omitted. $\qquad\square$

**Lemma 7.** *With $0 < \tau \leq \frac{1}{\max\{1, 2\ln(A/c)\}}$, for any $I \in \mathcal{I}$ and $h \in [H]$, we have:*

$$
\max\{\|\ln \pi_t^h(\cdot|I)\|_\infty, \|\ln \pi_\mu^h(\cdot|I)\|_\infty\} \leq \frac{2H}{\tau},
$$

$$
\|\pi_{t+1}^h(\cdot|I) - \pi_t^h(\cdot|I)\|_1 \leq \frac{7}{2}\eta H.
$$

*Proof.* The updated strategy of RegFTRL satisfies:

$$
\pi_{t+1}^h(a|I) \propto \exp\left\{(1 - \eta\tau)\ln \pi_t^h(a|I) + \eta\tau \left[Q_t^{h,\tau}(I, a) + \tau \ln \mu^h(a|I)\right]/\tau\right\}
$$

Let us denote $w_{t+1}^{h,\tau}(I, a) := (1 - \eta\tau)0.w_{t,\tau}^h(I, a) + \eta\tau \left[Q_t^{h,\tau}(I, a) + \tau \ln \mu^h(a|I)\right]$, and thus $\pi_t^h(a|I) \propto \exp\{w_t^{h,\tau}(I, a)/\tau\}$. The bounds of $w_t^{h,\tau}(I, a)$ can be obtained by a induction and lemma 6:

$$
(h - H - 1)\tau \ln \frac{A}{c} \leq w_t^{h,\tau}(I, a) \leq (H - h + 1) + \mathbf{1}_{h \neq H} \cdot (H - h - 1)\tau \ln \frac{A}{c}, \forall h, t, I, a.
$$

Therefore, for any $a_1, a_2 \in \mathcal{A}(\mathcal{I})$, we have

$$
\frac{\pi_t^h(a_1|I)}{\pi_t^h(a_2|I)} = \exp\left(\frac{w_t^{h,\tau}(I, a_1) - w_t^{h,\tau}(I, a_2)}{\tau}\right)
$$

$$
\leq \exp\left(\frac{(H - h + 1) + (2H - 2h + 1)\tau \ln \frac{A}{c}}{\tau}\right)
$$

Then,

$$\min_a \pi_t^h(a|I) \geq \frac{1}{A \cdot \exp\left(\frac{(H-h+1)+(2H-2h+1)\tau \ln \frac{A}{c}}{\tau}\right)} \sum_b \pi_t^h(b|I)$$

$$= \frac{1}{A \cdot \exp\left(\frac{(H-h+1)+(2H-2h+1)\tau \ln \frac{A}{c}}{\tau}\right)},$$

which gives

$$\|\ln \pi_t^h(\cdot|I)\|_\infty \leq \ln A + \frac{(H-h+1)+(2H-2h+1)\tau \ln \frac{A}{c}}{\tau}$$

$$\leq \frac{(H-h+1)+2(H-h+1)\tau \ln \frac{A}{c}}{\tau}$$

$$= \frac{H-h+1}{\tau}(1 + 2\tau \ln \frac{A}{c}) \leq 2\frac{H-h+1}{\tau} \leq 2\frac{H}{\tau}$$

The bound of $\|\ln \pi_\mu^h(\cdot|I)\|_\infty$ can be proven similarly and is therefore omitted.

Next, we prove the second statement. We invoke the following lemma first.

**Lemma 8** (Lemma 24, Mei et al. (2020)). *Let $\pi, \pi' \in \Delta_\mathcal{A}$ such that $\pi(a) \propto \exp(\theta(a)), \pi'(a) \propto \exp(\theta'(a))$ for some $\theta, \theta' \in \mathbb{R}^A$. It holds that*

$$\|\pi - \pi'\|_1 \leq \|\theta - \theta'\|_\infty.$$

With this lemma, for any $t \geq 0$, we have

$$\|\pi_{t+1}^h(\cdot|I) - \pi_t^h(\cdot|I)\|_1$$

$$\leq \min_{c \in \mathbb{R}} \|\ln \pi_{t+1}^h(\cdot|I) - \ln \pi_t^h(\cdot|I) - c \cdot \mathbf{1}\|_\infty$$

$$\leq \eta \|Q_t^{h,\tau}(I, \cdot)\|_\infty + \eta\tau \|\ln \pi_t^h(\cdot|I)\|_\infty + \eta\tau \|\ln \mu^h(\cdot|I)\|_\infty$$

$$\leq \eta\left((H-h+1)+(H-h)\tau \ln \frac{A}{c}\right) + \eta(H-h+1)(1 + 2\tau \ln \frac{A}{c}) + \eta\tau \ln \frac{A}{c}$$

$$\leq \eta(H-h+1)\left(2 + 3\tau \ln \frac{A}{c}\right) \leq \frac{7}{2}\eta(H-h+1) \leq \frac{7}{2}\eta H.$$

Thus, the proof is completed. $\qquad\qquad\square$

**Lemma 9.** *For $0 \leq t_1 < t_2, h \in [H], I \in \mathcal{I}$, we have*

$$D_{KL}^I(\pi_\mu^h, \pi_{t_2}^h) + \eta\tau D_{KL}^I(\pi_{t_2}^h, \pi_\mu^h) \leq (1 - \eta\tau)^{t_2 - t_1}\left(D_{KL}^I(\pi_\mu^h, \pi_{t_1}^h) + \eta\tau D_{KL}^I(\pi_{t_1}^h, \pi_\mu^h)\right)$$

$$+ 2\eta \sum_{l=t_1}^{t_2-1}(1 - \eta\tau)^{t_2-1-l}\|Q_l^{h,\tau}(I, \cdot) - Q_{\pi_\mu}^{h,\tau}(I, \cdot)\|_\infty.$$

*Proof.* From the update rule of RegFTRL, we have

$$\begin{cases} \ln \pi_{t+1}^h - (1 - \eta\tau) \ln \pi_t^h \overset{1}{=} \eta\left(Q_t^{h,\tau} + \eta \ln \mu^h\right), I \in \mathcal{I}_1 \\ \ln \pi_{t+1}^h - (1 - \eta\tau) \ln \pi_t^h \overset{1}{=} \eta\left(- Q_t^{h,\tau} + \eta \ln \mu^h\right), I \in \mathcal{I}_2 \end{cases} \tag{8}$$

$$\begin{cases} \eta\tau \ln \pi_\mu^h \overset{1}{=} \eta\left(Q_{\pi_\mu}^{h,\tau} + \eta \ln \mu^h\right), I \in \mathcal{I}_1 \\ \eta\tau \ln \pi_\mu^h \overset{1}{=} \eta\left(- Q_{\pi_\mu}^{h,\tau} + \eta \ln \mu^h\right), I \in \mathcal{I}_2 \end{cases} \tag{9}$$

Here $x \overset{1}{=} y$ denotes $x = y + c \cdot \mathbf{1}$ for some constant $c \in \mathbb{R}$. Subtracting (8) from (9) and taking inner product with $\pi_{t+1}^h - \pi_\mu^h$ gives

$$\langle \ln \pi_{t+1}^h - (1 - \eta\tau) \ln \pi_t^h - \eta\tau \ln \pi_\mu^h, \pi_{t+1}^h - \pi_\mu^h \rangle = \eta\delta(I)\langle Q_t^{h,\tau} - Q_{\pi_\mu}^{h,\tau}, \pi_{t+1}^h - \pi_\mu^h \rangle$$

$$\leq 2\eta \|Q_t^{h,\tau} - Q_{\pi_\mu}^{h,\tau}\|_\infty.$$

On the other hand, the LHS of the above inequality can be re-written as

$$\text{(LHS)} = D_{\text{KL}}^I(\pi_\mu^h, \pi_{t+1}^h) - (1 - \eta\tau)D_{\text{KL}}^I(\pi_\mu^h, \pi_t^h) + (1 - \eta\tau)D_{\text{KL}}^I(\pi_{t+1}^h, \pi_t^h) + \eta\tau D_{\text{KL}}^I(\pi_{t+1}^h, \pi_\mu^h).$$

Therefore, we have

$$D_{\text{KL}}^I(\pi_\mu^h, \pi_{t+1}^h) + \eta\tau D_{\text{KL}}^I(\pi_{t+1}^h, \pi_\mu^h) \leq (1 - \eta\tau)D_{\text{KL}}^I(\pi_\mu^h, \pi_t^h) + 2\eta\|Q_t^{h,\tau} - Q_{\pi_\mu}^{h,\tau}\|_\infty,$$

which gives:

$$D_{\text{KL}}^I(\pi_\mu^h, \pi_{t_2}^h) + \eta\tau D_{\text{KL}}^I(\pi_{t_2}^h, \pi_\mu^h) \leq (1 - \eta\tau)^{t_2 - t_1}\big(D_{\text{KL}}^I(\pi_\mu^h, \pi_{t_1}^h) + \eta\tau D_{\text{KL}}^I(\pi_{t_1}^h, \pi_\mu^h)\big)$$
$$+ 2\eta \sum_{l=t_1}^{t_2-1}(1 - \eta\tau)^{t_2-1-l}\|Q_l^{h,\tau}(I, \cdot) - Q_{\pi_\mu}^{h,\tau}(I, \cdot)\|_\infty$$

for $0 \leq t_1 < t_2$.

Thus, the proof is completed. $\qquad\square$

**Lemma 10.** *With* $0 < \eta\tau \leq \frac{2}{3}$ *and* $\alpha_t = \eta\tau$, *for any* $0 \leq t_1 < t_2$, $2 \leq h \leq H$, $I \in \mathcal{I}$, *we have*

$$|Q_{t_2}^{h-1,\tau}(I, a) - Q_{\pi_\mu}^{h-1,\tau}(I, a)| \leq (1 - \eta\tau)^{t_2-t_1}\Big[2H + 2\tau\mathbb{E}_{I'=I(ha),h\in I}\big[D_{KL}^{I'}(\pi_\mu, \pi_{t_1-1})\big]\Big]$$
$$+ 19\eta\tau \sum_{l=t_1-1}^{t_2-1}(1 - \eta\tau)^{t_2-1-l}\mathbb{E}_{I'=I(ha),h\in I}\big[\|Q_l^{h,\tau}(I', \cdot) - Q_{\pi_\mu}^{h,\tau}(I', \cdot)\|_\infty\big].$$

*Proof.* For $t_2 > t_1 \geq 0$, we have

$$Q_{t_2}^{h-1,\tau}(I, a) - Q_{\pi_\mu}^{h-1,\tau}(I, a)$$
$$= \mathbb{E}_{I'=I(ha),h\in I}\big[V_{t_2-1}^{h,\tau}(I') - V_{\pi_\mu}^{h,\tau}(I')\big]$$
$$= \mathbb{E}_{I'=I(ha),h\in I}\Big[(1 - \eta\tau)^{t_2-t_1}\big(V_{t_1-1}^{h,\tau}(I') - V_{\pi_\mu}^{h,\tau}(I')\big)$$
$$+ \eta\tau \sum_{l=t_1}^{t_2-1}(1 - \eta\tau)^{t_2-1-l}\big(f_{I'}^h(Q_l, \pi_l) - f_{I'}^h(Q_{\pi_\mu}, \pi_\mu)\big)\Big]$$
$$\leq (1 - \eta\tau)^{t_2-t_1}2H + \eta\tau\mathbb{E}_{I'=I(ha),h\in I}\Big[\sum_{l=t_1}^{t_2-1}(1 - \eta\tau)^{t_2-1-l}\big(f_{I'}^h(Q_l, \pi_l) - f_{I'}^h(Q_{\pi_\mu}, \pi_\mu)\big)\Big],$$

where $f_I^h(Q, \pi) := \langle\pi(\cdot|I), Q^{h,\tau}(I, \cdot)\rangle - \delta(I)\tau D_{\text{KL}}^I(\pi, \mu)$.

Denote $\hat{\pi}_t(\cdot|I) := \mathbf{1}_{\mathcal{P}(I)=1}\pi_t(\cdot|I) + \mathbf{1}_{\mathcal{P}(I)=2}\pi_\mu(\cdot|I)$. Then,

$$f_I^h(Q_t, \hat{\pi}_t) - f_I^h(Q_{\pi_\mu}, \hat{\pi}_t) = \langle Q_t^{h,\tau}(I, \cdot) - Q_{\pi_\mu}^{h,\tau}(I, \cdot), \hat{\pi}_t(\cdot|I)\rangle \leq \|Q_t^{h,\tau}(I, \cdot) - Q_{\pi_\mu}^{h,\tau}(I, \cdot)\|_\infty.$$
$$f_I^h(Q_{\pi_\mu}, \hat{\pi}_t) - f_I^h(Q_{\pi_\mu}, \pi_\mu) \leq 0.$$

Therefore, we have

$$f_I^h(Q_t, \pi_t) - f_I^h(Q_{\pi_\mu}, \pi_\mu)$$
$$\leq f_I^h(Q_t, \pi_t) - f_I^h(Q_t, \hat{\pi}_t) + \|Q_t^{h,\tau}(I, \cdot) - Q_{\pi_\mu}^{h,\tau}(I, \cdot)\|_\infty.$$

We here introduce a useful lemma first.

**Lemma 11** (Lemma 16, Cen et al. (2023)). *Let* $x \in \Delta_\mathcal{A}$ *be defined as*

$$x(a) \propto y(a)^{1-\eta\tau}\exp(-\eta w(a))$$

*for some* $w \in \mathbb{R}^A$ *and* $y \in \Delta_\mathcal{A}$. *It holds for all* $z \in \Delta_\mathcal{A}$ *that*

$$\frac{\eta}{1-\eta\tau}\big[\langle x - z, w\rangle - \tau\mathcal{H}(x) + \tau\mathcal{H}(z)\big] = D_{KL}(z, y) - \frac{1}{1-\eta\tau}D_{KL}(z, x) - D_{KL}(x, y).$$

With this lemma, for $I \in \mathcal{I}_2$, we have

$$f_I^h(Q_t, \pi_t) - f_I^h(Q_t, \hat{\pi}_t)$$
$$=\langle \pi_t(\cdot|I) - \pi_\mu(\cdot|I), Q_t^{h,\tau}(I, \cdot) - \tau \ln \mu(\cdot|I) \rangle - \tau \mathcal{H}^I(\pi_t) + \tau \mathcal{H}^I(\pi_\mu)$$
$$=\langle \pi_t(\cdot|I) - \pi_\mu(\cdot|I), Q_{t-1}^{h,\tau}(I, \cdot) - \tau \ln \mu(\cdot|I) \rangle - \tau \mathcal{H}^I(\pi_t) + \tau \mathcal{H}^I(\pi_\mu)$$
$$+ \langle \pi_t(\cdot|I) - \pi_\mu(\cdot|I), Q_t^{h,\tau}(I, \cdot) - Q_{t-1}^{h,\tau}(I, \cdot) \rangle$$
$$\leq \frac{1-\eta\tau}{\eta} D_{\mathrm{KL}}^I(\pi_\mu, \pi_{t-1}) - \frac{1}{\eta} D_{\mathrm{KL}}^I(\pi_\mu, \pi_t) - \frac{1-\eta\tau}{\eta} D_{\mathrm{KL}}^I(\pi_t, \pi_{t-1})$$
$$+ 2\|Q_t^{h,\tau}(I, \cdot) - Q_{t-1}^{h,\tau}(I, \cdot)\|_\infty,$$

and $f_I^h(Q_t, \pi_t) - f_I^h(Q_t, \hat{\pi}_t) = 0$ on $I \in \mathcal{I}_1$. Therefore, we have

$$f_I^h(Q_t, \pi_t) - f_I^h(Q_{\pi_\mu}, \pi_\mu)$$
$$\leq \|Q_t^{h,\tau}(I, \cdot) - Q_{\pi_\mu}^{h,\tau}(I, \cdot)\|_\infty + \mathbf{1}_{I \in \mathcal{I}_2} \cdot \Big( 2\|Q_t^{h,\tau}(I, \cdot) - Q_{t-1}^{h,\tau}(I, \cdot)\|_\infty$$
$$+ \frac{1-\eta\tau}{\eta} D_{\mathrm{KL}}^I(\pi_\mu, \pi_{t-1}) - \frac{1}{\eta} D_{\mathrm{KL}}^I(\pi_\mu, \pi_t) - \frac{1-\eta\tau}{\eta} D_{\mathrm{KL}}^I(\pi_t, \pi_{t-1}) \Big) \qquad (10)$$

By a similar argument,

$$f_I^h(Q_{\pi_\mu}, \pi_\mu) - f_I^h(Q_t, \pi_t)$$
$$\leq \|Q_t^{h,\tau}(I, \cdot) - Q_{\pi_\mu}^{h,\tau}(I, \cdot)\|_\infty + \mathbf{1}_{I \in \mathcal{I}_1} \cdot \Big( 2\|Q_t^{h,\tau}(I, \cdot) - Q_{t-1}^{h,\tau}(I, \cdot)\|_\infty$$
$$+ \frac{1-\eta\tau}{\eta} D_{\mathrm{KL}}^I(\pi_\mu, \pi_{t-1}) - \frac{1}{\eta} D_{\mathrm{KL}}^I(\pi_\mu, \pi_t) - \frac{1-\eta\tau}{\eta} D_{\mathrm{KL}}^I(\pi_t, \pi_{t-1}) \Big) \qquad (11)$$

Combining $(10) + \frac{1}{2}(11)$ gives

$$\frac{1}{2}\big(f_I^h(Q_t, \pi_t) - f_I^h(Q_{\pi_\mu}, \pi_\mu)\big)$$
$$\leq \frac{3}{2}\|Q_t^{h,\tau}(I, \cdot) - Q_{\pi_\mu}^{h,\tau}(I, \cdot)\|_\infty + 2\|Q_t^{h,\tau}(I, \cdot) - Q_{t-1}^{h,\tau}(I, \cdot)\|_\infty$$
$$+ \frac{1-\eta\tau}{\eta} G_{t-1}^h(I) - \frac{1}{\eta} G_t^h(I),$$

where $G_t^h(I) := \mathbf{1}_{I \in \mathcal{I}_2} \cdot D_{\mathrm{KL}}^I(\pi_\mu, \pi_t) + \mathbf{1}_{I \in \mathcal{I}_1} \cdot \frac{1}{2} D_{\mathrm{KL}}^I(\pi_\mu, \pi_t)$. Therefore, we have

$$Q_{t_2}^{h-1,\tau}(I, a) - Q_{\pi_\mu}^{h-1,\tau}(I, a)$$

$$\leq (1-\eta\tau)^{t_2-t_1} 2H + \eta\tau \mathbb{E}_{I'=I(ha), h \in I}\Big[ \sum_{l=t_1}^{t_2-1} (1-\eta\tau)^{t_2-1-l}\big(f_{I'}^h(Q_l, \pi_l) - f_{I'}^h(Q_{\pi_\mu}, \pi_\mu)\big)\Big]$$

$$\leq (1-\eta\tau)^{t_2-t_1} 2H + \eta\tau \mathbb{E}_{I'=I(ha), h \in I}\Big[ \sum_{l=t_1}^{t_2-1} (1-\eta\tau)^{t_2-1-l}\big(3\|Q_l^{h,\tau}(I', \cdot) - Q_{\pi_\mu}^{h,\tau}(I', \cdot)\|_\infty$$

$$+ 4\|Q_l^{h,\tau}(I', \cdot) - Q_{l-1}^{h',\tau}(I, \cdot)\|_\infty\big)\Big] + 2\tau(1-\eta\tau)^{t_2-t_1}\mathbb{E}_{I'=I(ha), h \in I}\big[G_{t_1-1}^h(I')\big]$$

$$\leq (1-\eta\tau)^{t_2-t_1} 2H + \eta\tau \mathbb{E}_{I'=I(ha), h \in I}\Big[ \sum_{l=t_1}^{t_2-1} (1-\eta\tau)^{t_2-1-l}\big(7\|Q_l^{h,\tau}(I', \cdot) - Q_{\pi_\mu}^{h,\tau}(I', \cdot)\|_\infty$$

$$+ 4\|Q_{l-1}^{h,\tau}(I', \cdot) - Q_{\pi_\mu}^{h,\tau}(I', \cdot)\|_\infty\big)\Big] + 2\tau(1-\eta\tau)^{t_2-t_1}\mathbb{E}_{I'=I(ha), h \in I}\big[G_{t_1-1}^h(I')\big]$$

$$\overset{\eta\tau \leq 2/3}{\leq} (1-\eta\tau)^{t_2-t_1} 2H + 19\eta\tau \mathbb{E}_{I'=I(ha), h \in I}\Big[ \sum_{l=t_1-1}^{t_2-1} (1-\eta\tau)^{t_2-1-l}\|Q_l^{h,\tau}(I', \cdot) - Q_{\pi_\mu}^{h,\tau}(I', \cdot)\|_\infty\Big]$$

$$+ 2\tau(1-\eta\tau)^{t_2-t_1}\mathbb{E}_{I'=I(ha), h \in I}\big[D_{\mathrm{KL}}^{I'}(\pi_\mu, \pi_{t_1-1})\big].$$

The other side can be shown with a similar proof and is therefore omitted. $\qquad \square$

**Lemma 12.** *It holds for all $h \in [H]$, $I \in \mathcal{I}$ and $\pi, \hat{\pi}$ that*

$$\max_{\hat{\pi}} \left( f_I^h(Q_{\pi_\mu}, (\hat{\pi}^1, \pi^2)) - f_I^h(Q_{\pi_\mu}, (\pi^1, \hat{\pi}^2)) \right) \leq \tau D_{KL}^I(\pi, \pi_\mu).$$

*Proof.* First, we have

$$
\begin{aligned}
&f_I^h(Q_{\pi_\mu}, (\hat{\pi}^1, \pi^2)) - f_I^h(Q_{\pi_\mu}, (\pi^1, \hat{\pi}^2)) \\
=& f_I^h(Q_{\pi_\mu}, (\hat{\pi}^1, \pi^2)) - f_I^h(Q_{\pi_\mu}, (\hat{\pi}^1, \pi_\mu^2)) \\
&- f_I^h(Q_{\pi_\mu}, (\pi^1, \hat{\pi}^2)) + f_I^h(Q_{\pi_\mu}, (\pi_\mu^1, \hat{\pi}^2)) \\
&- f_I^h(Q_{\pi_\mu}, (\pi_\mu^1, \hat{\pi}^2)) + f_I^h(Q_{\pi_\mu}, (\hat{\pi}^1, \pi_\mu^2)).
\end{aligned}
$$

Observe that for $I \in \mathcal{I}_1$,

$$
\begin{aligned}
&f_I^h(Q_{\pi_\mu}, (\pi_\mu^1, \hat{\pi}^2)) - f_I^h(Q_{\pi_\mu}, (\hat{\pi}^1, \pi_\mu^2)) = \langle \pi_\mu - \hat{\pi}, Q_{\pi_\mu} \rangle - \tau D_{KL}^I(\pi_\mu, \mu) + \tau D_{KL}^I(\hat{\pi}, \mu) \\
=& \tau \langle \pi_\mu - \hat{\pi}, \ln \frac{\pi_\mu}{\mu} \rangle - \tau D_{KL}^I(\pi_\mu, \mu) + \tau D_{KL}^I(\hat{\pi}, \mu) = \tau D_{KL}^I(\hat{\pi}, \pi_\mu).
\end{aligned}
$$

Similarly, for $I \in \mathcal{I}_2$, $f_I^h(Q_{\pi_\mu}, (\pi_\mu^1, \hat{\pi}^2)) - f_I^h(Q_{\pi_\mu}, (\hat{\pi}^1, \pi_\mu^2)) = \tau D_{KL}^I(\hat{\pi}, \pi_\mu)$. On the other hand, we can observe that for $I \in \mathcal{I}$,

$$
\begin{aligned}
f_I^h(Q_{\pi_\mu}, (\hat{\pi}^1, \pi^2)) - f_I^h(Q_{\pi_\mu}, (\hat{\pi}^1, \pi_\mu^2)) &= f_I^h(Q_{\pi_\mu}, (\pi_\mu^1, \pi^2)) - f_I^h(Q_{\pi_\mu}, \pi_\mu), \\
-f_I^h(Q_{\pi_\mu}, (\pi^1, \hat{\pi}^2)) + f_I^h(Q_{\pi_\mu}, (\pi_\mu^1, \hat{\pi}^2)) &= -f_I^h(Q_{\pi_\mu}, (\pi^1, \pi_\mu^2)) + f_I^h(Q_{\pi_\mu}, \pi_\mu).
\end{aligned}
$$

Therefore, we have

$$
\begin{aligned}
&f_I^h(Q_{\pi_\mu}, (\hat{\pi}^1, \pi^2)) - f_I^h(Q_{\pi_\mu}, (\pi^1, \hat{\pi}^2)) \\
=& f_I^h(Q_{\pi_\mu}, (\pi_\mu^1, \pi^2)) - f_I^h(Q_{\pi_\mu}, \pi_\mu) - f_I^h(Q_{\pi_\mu}, (\pi^1, \pi_\mu^2)) + f_I^h(Q_{\pi_\mu}, \pi_\mu) - \tau D_{KL}^I(\hat{\pi}, \pi_\mu) \\
=& \tau D_{KL}^I(\pi, \pi_\mu) - \tau D_{KL}^I(\pi, \pi_\mu) \leq \tau D_{KL}^I(\pi, \pi_\mu).
\end{aligned}
$$

Thus, the proof is completed. $\square$

**Theorem 6.** *Let $g_\mu^I(\pi) = D_{KL}^I(\pi, \mu) := D_{KL}(\pi(\cdot|I), \mu(\cdot|I))$, $\mu \geq \frac{c}{|\mathcal{A}|}$, $c \in (0, 1]$. With $0 < \eta\tau \leq \frac{2}{3}$, $0 < \tau \leq \frac{1}{\max\{1, 2\ln(|\mathcal{A}|/c)\}}$, and $\alpha_t = \eta\tau$, Algorithm 1 satisfies:*

$$\max_{I,a} |Q_{\pi_\mu}^{h,\tau} - Q_t^{h,\tau}| \leq (1 - \eta\tau)^{t-T_h} t^{H-h}, \tag{12}$$

$$\mathcal{E}_\tau(\pi_t) \leq 8(1 - \eta\tau)^{t-T_1} \left( \frac{5}{3}H + 3t^H \right), \tag{13}$$

*where $t \geq T_h := (H - h)T_s$, $T_s = \lceil \frac{1}{\eta\tau} \ln 2(9H + 19) \rceil$.*

*Proof.* We prove Theorem 6 by induction. By definition, we have $\|Q_{\pi_\mu}^{H,\tau} - Q_0^{H,\tau}\|_\infty = \|Q_{\pi_\mu}^{H,\tau}\|_\infty \leq 1$, and $\|Q_{\pi_\mu}^{H,\tau} - Q_t^{H,\tau}\|_\infty = \|r^H - r^H\|_\infty = 0$ for $t \geq 0$. So (12) holds for $h = H$. When (12) holds for some $h$, we can invoke lemma 10 with $t_1 = T_h + 1$ and $t_2 = t \geq T_{h-1}$, which yields

$$
\begin{aligned}
&\|Q_t^{h-1,\tau}(I, \cdot) - Q_{\pi_\mu}^{h-1,\tau}(I, \cdot)\|_\infty \leq (1 - \eta\tau)^{t-T_h-1} \left[ 2H + 2\tau \mathbb{E}_{I'=I(ha), h \in I} \left[ D_{KL}^{I'}(\pi_\mu, \pi_{T_h}) \right] \right] \\
&+ 19\eta\tau \sum_{l=T_h}^{t-1} (1 - \eta\tau)^{t-1-l} \mathbb{E}_{I'=I(ha), h \in I} \left[ \|Q_l^{h,\tau}(I', \cdot) - Q_{\pi_\mu}^{h,\tau}(I', \cdot)\|_\infty \right] \\
&\leq (1 - \eta\tau)^{t-T_h-1} \left[ 2H + 4H \right] + 19\eta\tau \sum_{l=T_h}^{t-1} (1 - \eta\tau)^{t-T_h-1} l^{H-h} \\
&\leq (1 - \eta\tau)^{t-T_h-1} (1 - \eta\tau)^{T_s-1} \left[ 6H + 19\eta\tau t^{H-h+1} \right].
\end{aligned}
$$

The second inequality follows from the fact that

$$\tau D_{\mathrm{KL}}^I(\pi_\mu, \pi_t) \le \tau \|\ln \pi_\mu(\cdot|I) - \ln \pi_t(\cdot|I)\|_\infty \le \tau \max\{\|\ln \pi_\mu(\cdot|I)\|_\infty, \|\ln \pi_t(\cdot|I)\|_\infty\}$$
$$\le 2H \quad \text{by lemma 7.}$$

Therefore, with $T_s = \lceil \frac{1}{\eta\tau} \ln 2(9H+19) \rceil$, we have

$$\|Q_t^{h-1,\tau} - Q_{\pi_\mu}^{h-1,\tau}\|_\infty \le (1-\eta\tau)^{t-T_{h-1}} t^{H-h+1}.$$

This completes the proof of (12).

Next, we prove the second statement. Let us denote $\hat\pi_t(\cdot|I) = \mathbf{1}_{I \in \mathcal{I}_1} \cdot \pi(\cdot|I) + \mathbf{1}_{I \in \mathcal{I}_2} \cdot \pi_t(\cdot|I)$ and $\bar\pi_t(\cdot|I) = \mathbf{1}_{I \in \mathcal{I}_1} \cdot \pi_t(\cdot|I) + \mathbf{1}_{I \in \mathcal{I}_2} \cdot \pi(\cdot|I)$. Then,

$$V_{\hat\pi_t}^{h-1,\tau}(I) - V_{\pi_\mu}^{h-1,\tau}(I)$$
$$= \langle \hat\pi_t(I), Q_{\hat\pi_t}^{h-1,\tau}(I) \rangle - \delta(I)\tau D_{\mathrm{KL}}^I(\hat\pi_t, \mu) - \langle \pi_\mu(I), Q_{\pi_\mu}^{h-1,\tau}(I) \rangle + \delta(I)\tau D_{\mathrm{KL}}^I(\pi_\mu, \mu)$$
$$= f_I^{h-1}(Q_{\pi_\mu}, \hat\pi_t) - f_I^{h-1}(Q_{\pi_\mu}, \pi_\mu) + \langle \hat\pi_t(I), Q_{\hat\pi_t}^{h-1,\tau}(I) - Q_{\pi_\mu}^{h-1,\tau}(I) \rangle$$
$$\le f_I^{h-1}(Q_{\pi_\mu}, \hat\pi_t) - f_I^{h-1}(Q_{\pi_\mu}, \mathbf{1}_{I \in \mathcal{I}_1} \cdot \pi_t + \mathbf{1}_{I \in \mathcal{I}_2} \cdot \pi_\mu) + \max_{I'} \left[ V_{\hat\pi_t}^{h,\tau}(I') - V_{\pi_\mu}^{h,\tau}(I') \right]$$
$$\le \max_\pi \left( f_I^{h-1}(Q_{\pi_\mu}, \hat\pi_t) - f_I^{h-1}(Q_{\pi_\mu}, \bar\pi_t) \right) + \max_{I'} \left[ V_{\hat\pi_t}^{h,\tau}(I') - V_{\pi_\mu}^{h,\tau}(I') \right]$$

By lemma 12 and lemma 9 with $t_2 = t, t_1 = T_h$, we have

$$\max_\pi \left( f_I^h(Q_{\pi_\mu}, \hat\pi_t) - f_I^h(Q_{\pi_\mu}, \bar\pi_t) \right) \le \tau D_{\mathrm{KL}}^I(\pi_t, \pi_\mu)$$

$$\le \frac{1}{\eta} \left[ D_{\mathrm{KL}}^I(\pi_\mu, \pi_t) + \eta\tau D_{\mathrm{KL}}^I(\pi_t, \pi_\mu) \right]$$

$$\le \frac{1}{\eta} \left[ (1-\eta\tau)^{t-T_h} \left( D_{\mathrm{KL}}^I(\pi_\mu, \pi_{T_h}) + \eta\tau D_{\mathrm{KL}}^I(\pi_{T_h}, \pi_\mu) \right) \right.$$

$$\left. + 2\eta \sum_{l=T_h}^{t-1} (1-\eta\tau)^{t-1-l} \|Q_l^{h,\tau}(I, \cdot) - Q_{\pi_\mu}^{h,\tau}(I, \cdot)\|_\infty \right]$$

$$\le \frac{1}{\eta} \left[ (1-\eta\tau)^{t-T_h} \left( \frac{2H}{\tau} + 2\eta H + 2\eta \sum_{l=T_h}^{t-1} (1-\eta\tau)^{-1} l^{H-h} \right) \right]$$

$$\le (1-\eta\tau)^{t-T_h} \left( \frac{10}{3} H + 6t^{H-h+1} \right). \tag{14}$$

Therefore, we have

$$V_{\hat\pi_t}^{h-1,\tau}(I) - V_{\pi_\mu}^{h-1,\tau}(I)$$
$$\le (1-\eta\tau)^{t-T_{h-1}} \left( \frac{10}{3} H + 6t^{H-h+2} \right) + \max_{I'} \left[ V_{\hat\pi_t}^{h,\tau}(I') - V_{\pi_\mu}^{h,\tau}(I') \right] \tag{15}$$

We next prove that the following results by induction.

$$\max_{I,\pi} \left( V_{\hat\pi_t}^{h,\tau}(I) - V_{\pi_\mu}^{h,\tau}(I) \right) \le 2(1-\eta\tau)^{t-T_h} \left( \frac{10}{3} H + 6t^{H-h+1} \right). \tag{16}$$

Since $V_{\hat\pi_t}^{H,\tau}(I) = f_I^H(r^H, \hat\pi_t) = f_I^H(Q_{\pi_\mu}, \hat\pi_t)$ and $V_{\bar\pi_t}^{H,\tau}(I) = f_I^H(r^H, \bar\pi_t) = f_I^H(Q_{\pi_\mu}, \bar\pi_t)$, the claim holds for $h = H$ by invoking (14). When the claim holds for some $2 \le h \le H$, by invoking 15, we have

$$\max_{\pi,I} \left( V_{\hat\pi_t}^{h-1,\tau}(I) - V_{\pi_\mu}^{h-1,\tau}(I) \right)$$

$$\le (1-\eta\tau)^{t-T_{h-1}} \left( \frac{10}{3} H + 6t^{H-h+2} \right) + \max_{\pi,I'} \left[ V_{\hat\pi_t}^{h,\tau}(I') - V_{\pi_\mu}^{h,\tau}(I') \right]$$

$$\le (1-\eta\tau)^{t-T_{h-1}} \left( \frac{10}{3} H + 6t^{H-h+2} \right) + 2(1-\eta\tau)^{t-T_h} \left( \frac{10}{3} H + 6t^{H-h+1} \right)$$

$$\le 2(1-\eta\tau)^{t-T_{h-1}} \left( \frac{10}{3} H + 6t^{H-h+2} \right).$$

Therefore the claim holds for any $h \in [H]$. We can prove the following claim by following a similar argument:

$$\max_{I,\pi} \left( V^{h,\tau}_{\pi_\mu}(I) - V^{h,\tau}_{\hat{\pi}_t}(I) \right) \leq 2(1-\eta\tau)^{t-T_h} \left( \frac{10}{3}H + 6t^{H-h+1} \right). \tag{17}$$

By combing (16) and (17), we have

$$\max_{\pi,I} \left( V^{h,\tau}_{\hat{\pi}_t}(I) - V^{h,\tau}_{\tilde{\pi}_t}(I) \right) = \max_{\pi,I} \left( V^{h,\tau}_{\hat{\pi}_t}(I) - V^{h,\tau}_{\pi_\mu}(I) + V^{h,\tau}_{\pi_\mu}(I) - V^{h,\tau}_{\tilde{\pi}_t}(I) \right)$$

$$\leq 8(1-\eta\tau)^{t-T_h} \left( \frac{5}{3}H + 3t^{H-h+1} \right). \tag{18}$$

Note that (18) is a stronger claim for (13), and thus (13) can be obtained by taking $h = H$:

$$\mathcal{E}_\tau(\pi_t) \leq 8(1-\eta\tau)^{t-T_1} \left( \frac{5}{3}H + 3t^H \right).$$

Thus, the proof is completed. $\qquad\square$

# B  PRACTICAL IMPLEMENTATION

## B.1  IMPLEMENTATION OF FOLLOWMU

In this section, we introduce the implementation of FollowMu. We employ the actor-critic framework to develop FollowMu due to its scalability (Sutton & Barto, 2018). Let $A(I, a; \theta_t)$ be the actor network parameterized by $\theta_t$, and $V(I; \omega_t)$ be the critic network parameterized by $\omega_t$. At the time step $t$, the critic network $V(I; \omega_t)$ is trained to approximate the value function $V_{\pi_t}(I)$ of the real-time strategy, and the actor network $A(I, a; \theta_t)$ is trained to fit the cumulative advantage function of past iterations plus the Q-function of the current-iterate strategy (with the regularized term):

$$A(I, a; \theta_t) \simeq \sum_{k=0}^{t-1} \left[ Q_{\pi_k}(I, a) - V_{\pi_k}(I) - \tau \log \frac{\pi_k(a|I)}{\mu(a|I)} \right] + Q_{\pi_t}(I, a) - \tau \log \frac{\pi_t(a|I)}{\mu(a|I)}$$

$$\simeq [A(I, a; \theta_{t-1}) - V(I; \omega_{t-1})] + G - \tau \log \frac{\pi_t(a|I)}{\mu(a|I)}, \tag{19}$$

where $G$ is the empirical estimator of $Q_{\pi_t}(I, a)$. Then, if we take $\psi$ to be the entropy regularizer, the next-iterate strategy can be computed by:

$$\pi_{t+1}(a|I) \propto \exp(z_t(I, a)), \quad z_t(I, a) \simeq A(I, a; \theta_t) - V(I; \omega_t). \tag{20}$$

Here we employ the advantage function $Q_{\pi_t}(I, a) - V_{\pi_t}(I)$, as a substitution for the Q-function $Q_{\pi_t}(I, a)$, a choice made for the sake of enhancing numerical stability and robustness. Despite this alteration, the strategy update formulation in the manner of Eq.(20) remains equivalent to the updated strategy employed in RegFTRL, attributed to the shift-invariant nature inherent in the softmax function. Meanwhile, the reference strategy will be updated $\mu \leftarrow \pi_t$ every $N$ iterations.

We summarize our implementation of FollowMu in Algorithm 1, where the return $G$ is estimated by Monte Carlo method, and the loss of actor and critic network are computed by MSE loss. Note that we use the clipped cumulative advantage function in practice:

$$A(I, a; \theta_t) = \min \left\{ \ell, \max \left\{ 0, A(I, a; \theta_{t-1}) - V(I; \omega_{t-1}) \right\} \right\} + G - \tau \log \frac{\pi_t(a|I)}{\mu(a|I)},$$

where the clipping operator is employed to ensure the stability of the training process, and $\ell > 0$ controls the strength of clipping. The clipping operator $\min\{\ell, x\}$ serves to effectively limit the magnitude of the cumulative advantage function, thereby preventing it from becoming excessively large and leading to performance collapse. Conversely, when dealing with cumulative values that are too small, we employ the positive clipping operator $\max\{0, x\}$ instead of $\max\{-\ell, x\}$ to truncate these values. This choice is based on empirical observations, as the positive clipping operator has faster convergence rate. In fact, this clipping operation is identical to the one used in CFR+ (Tammelin, 2014), which is a simple yet highly effective technique for improving performance (Bowling et al., 2017). Additionally, when collecting the buffer, the current policy will be perturbed by a small $\epsilon$ probability.

---

**Algorithm 1** FollowMu

---

Initialize $\pi_0$ as uniform, $\theta_0, \omega_0$ as arbitrary
**for** $t$ *in* $0, 1, \cdots$ **do**
    **if** $t \mod N = 0$ **then**
      |   $\mu \leftarrow \pi_t$
    **end**
    Collect replay buffer: $\mathcal{B}_t \sim \pi_t$
    **for** $k$ *in* $0, 1, \cdots$ **do**
      Fetch a mini-batch of samples $\mathcal{D}$ from the replay buffer $\mathcal{B}_t$
      **for** $(I, a) \in \mathcal{D}$ **do**
        $G \leftarrow \text{Return}(I, a, \mathcal{D})$
        **if** $t = 0$ **then**
        |   $A_{\text{tmp}} \leftarrow 0$
        **end**
        **else**
        |   $A_{\text{tmp}} \leftarrow \min\{\ell, \max\{0, A(I, a; \theta_{t-1}) - V(I; \omega_{t-1})\}\}$
        **end**
        $A_{\text{target}} \leftarrow A_{\text{tmp}} + G - \tau \log \frac{\pi_t(a|I)}{\mu(a|I)}$
        $\theta_t \leftarrow \text{UpdateActor}(I, a, A_{\text{target}})$
      **end**
      **for** $I \in \mathcal{D}$ **do**
        $G \leftarrow \text{Return}(I, \mathcal{D})$
        $\omega_t \leftarrow \text{UpdateCritic}(I, G)$
      **end**
    **end**
    $\pi_{t+1}(a|I) \propto \exp(A(I, a; \theta_t) - V(I; \omega_t))$
**end**

---

## B.2 EXPERIMENTAL SETTINGS

### B.2.1 FULL-INFORMATION FEEDBACK SETTING

The payoff matrices of M-NE from Wei et al. (2021) is as follows:

|       | $y_1$ | $y_2$ | $y_3$ | $y_4$ | $y_5$ |
|-------|-------|-------|-------|-------|-------|
| $x_1$ | 0     | 1     | −1    | 0     | 0     |
| $x_2$ | −1    | 0     | 1     | 0     | 0     |
| $x_3$ | 1     | −1    | 0     | 0     | 0     |
| $x_4$ | 1     | −1    | 0     | −2    | 1     |
| $x_5$ | 1     | −1    | 0     | 1     | −2    |

M-NE has the following set of Nash equilibria: $\Pi_*^1 = \{(1/3, 1/3, 1/3, 0, 0)\}, \Pi_*^2 = \{y \in \Delta^5 | y_1 = y_2 = y_3; y_5/2 \leq y_4 \leq 2y_5\}$. For the random utility game, the $50 \times 50$ payoff matrix is drawn from a standard Gaussian distribution in an i.i.d. manner. The benchmarks of Kuhn/Leduc Poker is from OpenSpiel. The hyper-parameters for RegFTRL in NFGs are listed in Table 2, and the hyper-parameters in EFGs are listed in Table 3.

Table 2: Hyper-Parameter Settings of RegFTRL in M-NE/Random Game.

|             | learning rate $\eta$ | regularization parameter $\tau$ | update period $N$ |
|-------------|----------------------|---------------------------------|-------------------|
| RegFTRL-A   | $\frac{10}{1+10\times\tau}, \frac{10}{1+10\times\tau}$ | $10^{1-t/500}, 10^{1-t/300}$ | $0, 0$ |
| RegFTRL-D   | $\frac{15}{1+15\times\tau}, \frac{6}{1+6\times\tau}$ | $\frac{15}{t}, \frac{6}{t}$ | $10, 100$ |
| M-RegFTRL-A | $10, 10$ | $10^{1-t/500}, 10^{1-t/300}$ | $0, 0$ |
| M-RegFTRL-D | $10, 6$ | $\frac{10}{t}, \frac{6}{t}$ | $10, 100$ |
| 2-RegFTRL-A | $10, 20$ | $10^{1-t/500}, 10^{1-t/300}$ | $0, 0$ |
| 2-RegFTRL-D | $10, 6.5$ | $\frac{10}{t}, \frac{6}{t}$ | $10, 100$ |

Table 3: Hyper-Parameter Settings of RegFTRL in Kuhn/Leduc Poker.

| | learning rate $\eta$ | regularization parameter $\tau$ | update period $N$ |
|---|---|---|---|
| RegFTRL-A | $\frac{1}{\sqrt{t}}, \frac{1}{\sqrt{t}}$ | $\frac{1}{\sqrt{t}}, \frac{5}{\sqrt{t}}$ | $0, 0$ |
| RegFTRL-D | $0.3, 0.11$ | $0.1, 1$ | $30, 30$ |
| M-RegFTRL-A | $\frac{1}{\sqrt{t}}, \frac{1}{\sqrt{t}}$ | $\frac{0.5}{\sqrt{t}}, \frac{3}{\sqrt{t}}$ | $0, 0$ |
| M-RegFTRL-D | $0.1, 0.11$ | $0.1, 1$ | $30, 30$ |
| 2-RegFTRL-A | $\frac{1}{\sqrt{t}}, \frac{1}{\sqrt{t}}$ | $\frac{2}{\sqrt{t}}, \frac{20}{\sqrt{t}}$ | $0, 0$ |
| 2-RegFTRL-D | $0.1, 0.11$ | $0.5, 7$ | $30, 30$ |

### B.2.2 NEURAL-BASED SAMPLE SETTING

The benchmarks of Kuhn/Leduc Poker and the implementation of NFSP are all from OpenSpiel, and all the experiments are run on A30. The hyper-parameters for FollowMu are listed in Table 4, while those for NFSP are listed in Table 5, which are referenced from the report (Walton & Lisý, 2021).

Table 4: Hyper-Parameter Settings of FollowMu in Kuhn/Leduc Poker.

| Parameter | Value |
|---|---|
| hidden_layers_sizes | $[128, 128]$ |
| batch_size | 1024 |
| mini_batch_size | 128/256 |
| logit_learning_rate | 0.001/0.0005 |
| critic_learning_rate | 0.005 |
| max_global_gradient_norm | 10.0 |
| optimizer_str | sgd |
| eta | 0.2 |
| refer_policy_update_every | 200/500 |
| clip_strength | 100 |

Table 5: Hyper-Parameter Settings of NFSP in Kuhn/Leduc Poker.

| Parameter | Value |
|---|---|
| hidden_layers_sizes | $[128, 128]$ |
| replay_buffer_capacity | 200000 |
| reservoir_buffer_capacity | 2000000 |
| min_buffer_size_to_learn | 1000 |
| anticipatory_param | 1 |
| batch_size | 128 |
| learn_every | 128 |
| rl_learning_rate | 0.01 |
| sl_learning_rate | 0.01 |
| optimizer_str | sgd |
| update_target_network_every | 19200 |
| discount_factor | 1.0 |
| epsilon_decay_duration | 10000000 |
| epsilon_start | 0.06 |
| epsilon_end | 0.001 |

## C  ADDITIONAL PRELIMINARIES

### C.1  GAME DECOMPOSITION

Several recent works have shown that an arbitrary game (normal-form type or differential-form type) can be uniquely decomposed into a sum of Hamiltonian and potential components through the generalized Helmholtz decomposition theorem (Balduzzi et al., 2018; Letcher et al., 2019). There are thus two "pure" games: Hamiltonian games (only the Hamiltonian component is present) and potential games (only the potential component). Hamiltonian games, such as Rock-Paper-Scissors, are actually divergence-free vector fields where the cyclic behaviors arise (Balduzzi et al., 2018). Hence, FTRL will get stuck in cycles around equilibrium if the Hamiltonian component of the underlying game is dominant. On the other hand, a game is a potential game if there is a single potential function $g$ such that $V_{\pi^1,\pi^2} - V_{\hat{\pi}^1,\pi^2} = -g(\pi^1,\pi^2) + g(\hat{\pi}^1,\pi^2)$ for all $\pi^1, \hat{\pi}^1, \pi^2$. Potential games are well-studied because they can be solved by following the gradient dynamics (Monderer & Shapley, 1996; Balduzzi et al., 2018).

### C.2  FOLLOW-THE-REGULARIZED-LEADER

FTRL is an intuitive algorithm: at each time step it maximizes the sum of the past returns with a regularization. For conciseness, we only present the definition of FTRL in NFGs here. Formally, FTRL dynamics is defined as follows:

$$\pi_t^i = \underset{p \in \Delta_{\mathcal{A}}}{\arg\max}[\eta\langle p, y_t^i\rangle - \psi_i(p)], \tag{21}$$

$$y_t^i(a) = \int_0^t \delta^i \cdot Q_{\pi_k}(a)dk, \quad \delta^i = 2 \cdot \mathbf{1}_{i=1} - 1,$$

where $\langle \cdot, \cdot \rangle$ means inner product, $\eta > 0$ is the learning rate, and the regularization function $\psi : \Delta_{\mathcal{A}} \to \mathbb{R}$ is strictly convex and continuously differentiable on $\Delta_{\mathcal{A}}$. Note that $\int_0^t Q_{\pi_k}(a)dk = \sum_{k=0}^{t-1} Q_{\pi_k}(a)$ under discrete-time settings.

Two prototypical examples of FTRL can be yielded by choosing different regularizers: 1) Replicator Dynamics (RD) induced by the entropy regularizer $\psi_i(p) = \sum_a p(a)\ln p(a)$; and 2) Projection Dynamics (PD) induced by the (square) Euclidean regularizer $\psi_i(p) = \frac{1}{2}\sum_a |p(a)|^2$ (Mertikopoulos et al., 2018).

RD is an important learning dynamics studied in evolution game theory (Hofbauer & Sigmund, 1998; Zeeman, 2006), where the central focus is to mimic the population's evolution process. The dynamics of RD can be given by the following differential equation:

$$\frac{d}{dt}\pi_t^i(a) = \pi_t^i(a)\delta^i(Q_{\pi_t}(a) - V_{\pi_t}). \tag{22}$$

PD is introduced as a geometric model of the evolution of play in population games (Friedman, 1991). Denoting the support set of policy as $\text{supp}(\pi_t^i) = \{a \in \mathcal{A} : \pi_t^i(a) > 0\}$, the dynamics of PD can be defined as follows:

$$\frac{d}{dt}\pi_t^i(a) = \delta^i Q_{\pi_t}(a) - |\text{supp}(\pi_t^i)|^{-1} \sum_{a' \in \text{supp}(\pi_t^i)} \delta^i Q_{\pi_t}(a'), \tag{23}$$

if $a \in \text{supp}(\pi_t^i)$, and $\frac{d}{dt}\pi_t^i(a) = 0$ otherwise.