# OpenReview forum: "Regularization is Enough for Last-Iterate Convergence in Zero-Sum Games"
_ICLR.cc/2024/Conference — Submitted to ICLR 2024_

### Official Review · Reviewer_onGj · 2023-10-30

**Soundness:** 3 good
**Presentation:** 3 good
**Contribution:** 2 fair
**Rating:** 5
**Confidence:** 4

**Summary:**

The authors mainly study a variant of FTRL that incorporates adaptive regularization (RegFTRL). They show that RegFTRL converges in a last-iterate sense to approximate Nash equilibria, and to exact Nash equilibria through the use of adaptive regularization. They also propose FollowMu, an implementation of RegFTRL that uses a neural network as a function approximator, for model-free reinforcement learning. Finally, they conduct experiments to support the theoretical findings.

**Strengths:**

The paper focuses on an important problem that has received considerable attention recently. Unlike much of prior work, the paper focuses on using (adaptive) regularization to guarantee last-iterate convergence, in lieu of using optimism or extra-gradients. Such approaches tend to perform well in practice, so any new theoretical insights about their behavior are definitely valuable. The presentation overall is reasonable, and the results appear to be sound.

**Weaknesses:**

The main issue pertains the novelty of the results. There are many existing papers with closely related results, such as 1) "Last-iterate
convergence with full- and noisy-information feedback in two-player zero-sum games;" 2) A unified approach to reinforcement learning, quantal response equilibria, and two-player zero-sum games;" 3) "Modeling strong and human-like gameplay with KL-regularized search;" and 4) an unpublished paper "No-Regret Learning in Strongly Monotone Games Converges to a Nash Equilibrium." Some of those papers are cited, but the discussion is inadequate, such as the comparison with magnetic mirror descent. Overall, it is known in the literature that in strongly monotone games algorithms such as FTRL exhibit last-iterate convergence, and so one can use adaptive regularization to extend such results in (non necessarily strongly) monotone games as well (by adding a strongly convex regularizer that makes the game strongly monotone). It is not clear to me how the new results are novel compared to the existing literature. Regarding the experimental evaluation (Section 5.1), experiments on very small games such as Kuhn or Leduc can be misleading, and it's hard to draw any definite conclusions. I would recommend using larger games.

**Questions:**

A couple of questions for the authors:

1. What is more concretely the problem of using optimism or extra-gradient in the context of Figure 1? You write that "...can impede
convergence, particularly when the real-time policy exhibits chaotic behavior," but I don't think I am following this.

2.  I am not sure I see the purpose of Section 2.3. Can the authors explain?

---

> ### Author Response · Authors · 2023-11-18
> **response to Reviewer onGj**
>
> Thanks for your comments which are quite helpful to further improve the quality of the paper.  We have made some necessary revisions (marked in blue font) in this version. Please refer to  the uploaded new version of submission and our point-to-piont responses below.
>
> **1. About novelty, and some of those papers are cited, but the discussion is inadequate.**
>
> We acknowledge that some of the discussion about related work was inadequate in the original submission, which is also argued by other reviewers. We have carefully addressed this issue and made some necessary modifications in the new version paper (see Section 1.1 on page 2~3). Please also refer to the responses to related comments raised by other reviewers.
>
> **2. Regarding the experimental evaluation (Section 5.1), experiments on very small games such as Kuhn or Leduc can be misleading, and it's hard to draw any definite conclusions. I would recommend using larger games.**
>
> In Section 5.1, our primary objective was to further validate our theoretical results, with a specific focus on convergence rather than scalability. To ensure a fair comparison with related works and for the sake of consistency with common experimental environments in the literature, we opted for the widely used Kuhn or Leduc game settings. It is true that these games have relatively small sizes, but they still capture the complexity of sequential decision-making, making them suitable for our convergence-focused analysis.
>
> Additionally, we acknowledge the importance of demonstrating the scalability potential of our proposed approach. In Section 5.2, we have included experiments with a larger game, Phantom Tic-Tac-Toe, to showcase the scalability of our approach.
>
> **3. What is more concretely the problem of using optimism or extra-gradient in the context of Figure 1? You write that "...can impede convergence, particularly when the real-time policy exhibits chaotic behavior," but I don't think I am following this.**
>
> In Figure 1, the primary focus is on illustrating the guidance of second-order information in dynamic settings for different types of games. Specifically:
>
> 1. In a Hamiltonian game, second-order information points toward the NE, facilitating convergence. The introduction of second-order information aids in convergence in this context.
> 2. Conversely, in a potential game, second-order information diverges from the NE. In such cases, introducing second-order information can decelerate convergence or even lead to divergence.
>
> The key reason for this divergence in potential games lies in the fact that the computation of second-order information depends on the gradient flow. In contrast, regularization, which is not dependent on the gradient flow, tends to provide more robust convergence performance.
>
> Furthermore, optimism or extra-gradient can be considered as first-order approximations of second-order information. Therefore, they may inherit similar issues in scenarios where second-order information introduces challenges.
>
> **4. I am not sure I see the purpose of Section 2.3. Can the authors explain?**
>
> The intention behind Section 2.3 is to illustrate that the introduction of regularization, as discussed in the paper, can be interpreted from the perspective of enhancing the potential component of a game. Specifically, it highlights that the regularization term in the objective function can be viewed as augmenting the potential component, which, in turn, facilitates the achievement of last-iterate convergence.
>
> In the revised version, we have moved Section 2.3 to Appendix C to ensure compliance with page limitations while maintaining the completeness of the content.

---

> > ### Comment · Reviewer_onGj · 2023-11-19
> > **Thank you for your Response**
> >
> > I thank the authors for their response. In your response you are claiming that "the implementation of the optimistic update approach often necessitates the computation of multiple strategies at each iteration, making it intricate and resource-intensive." I don't think I follow what you mean. What is intricate and resource intensive about the optimistic update? Optimism basically requires the same amount of compute compared to the vanilla variant.
> >
> > Moreover, I still haven't found any discussion about the paper "No-Regret Learning in Strongly Monotone Games Converges to a Nash Equilibrium." Can the authors elaborate on this?

---

> ### Author Response · Authors · 2023-11-20
> **response to Reviewer onGj**
>
> Thanks for your comment and we address your questions below.
>
> **About the optimistic update**
>
> Taking the update formula of OGDA from [1] as an example:
> $$
> \pi_{t}=\text{proj}[x_{t} - \eta F(\pi_{t-1})], x_{t+1}=\text{proj}[x_{t} - \eta F(\pi_t)],
> $$
> where $\text{proj}$ is the projection operator and $F(\pi)=Q_{\pi}$ is the gradient information. It can be observed that, compared to RegFTRL or the vanilla GDA, the optimistic update involves twice the amount of updates. Even in the unconstrained case, where the projection operator is removed, and the update formula can be rewritten as
> $$
> \pi_{t+1} = \pi_{t} - 2\eta F(\pi_{t}) + \eta F(\pi_{t-1}),
> $$
>
> it still requires twice the storage (i.e., in addition to storing $F(\pi_{t})$, it also needs to store the additional term $F(\pi_{t-1})$).
>
> Nevertheless, the original description may have been somewhat exaggerated, we acknowledge this and have modified it accordingly in the revised manuscript (see Section 1.1 on page 3).
>
> **Relation to [2]**
>
> [2] primarily explores the relationship between no-regret learning dynamics and time-average convergence to NE in strongly monotone games. While [2] doesn't specifically focus on last-iterate convergence, its Theorem 2 and Theorem 4 indeed establish that first-order and best response algorithms can achieve last-iterate convergence.
>
> - Theorem 2 in [2] shows that the first-order GDA algorithm, with a decreasing learning rate, achieves last-iterate convergence in strongly monotone games. This is equivalent to demonstrating that the GDA algorithm with regularization can converge in monotone games. In contrast, our algorithm provides convergence rates under a constant learning rate setting.
> - Theorem 4 in [2] establishes last-iterate convergence for best response algorithms. Although RegFTRL (as well as the optimistic/extra-gradient method) can be seen as aiming to approximate the implicit best-response problem, it remains distinct from explicit best response algorithms. Therefore, the analysis and proof techniques are different from [2].
>
> We acknowledge that [2] is related to our work, and we have added a discussion in the Related Work section (see Section 1.1 on page 3).
>
> [1] Linear Last-iterate Convergence in Constrained Saddle-point Optimization, Wei et al. ICLR 2021.
>
> [2] No-Regret Learning in Strongly Monotone Games Converges to a Nash Equilibrium. Unpublished paper 2022.

---

> > ### Comment · Reviewer_onGj · 2023-11-21
> > **Thank you for the Response**
> >
> > I thank the authors for their response. I have no further questions at the moment.

---

> > > ### Author Response · Authors · 2023-11-22
> > >
> > > Given the time constraints, and assuming we have sufficiently addressed the concerns you raised, would you consider re-evaluating our score? We sincerely appreciate your assistance and understanding.

---

### Official Review · Reviewer_3dkM · 2023-10-30

**Soundness:** 3 good
**Presentation:** 2 fair
**Contribution:** 2 fair
**Rating:** 5
**Confidence:** 4

**Summary:**

This paper focuses the problem of learning Nash equilibrium in two-player zero-sum games with last-iterate convergence guarantees. The authors proposed an algorithm called Regularized Follow the Regularized Leader (RegFRTL), which is variant of FTRL with adaptive regularization. For a fixed regularization, it is shown that RegFTRL has linear convergence rates to the unique Nash equilibrium of the regularized normal-form game (NFG) and extensive-form game (EFG). Moreover, by decreasing the regularization term or changing the referenece policy periodically (every $N$ iterations), it is proved that RegFTRL under entropy regularization converges to exact Nash equilibria (without a rate) in normal form two-player zero-sum game. Finally, the authors proposed an algorothm called FollowMu, which utilize the actor-critic framework parameterized by neural networks and empirical estimator of $Q$ function. Experimental results show fast convergence of RegFTRL and FollowMu in both NFGs and EFGs.

**Strengths:**

This paper focuses on an important problem of last-iterate convergence in games. The proposed approach is general for various regularization functions and has convergence results for both the normal-form games and extensive-form games. This paper is fairly well-written and easy to follow.

**Weaknesses:**

My main concerns are the novelty of the approach, and insufficient discussion on relation to previous works.

1. The proposed approach in this paper is very similar to the approach proposed in [1]. In [1], the authors proposed FTRL-SP and prove it has (1) linear last-iterate convergence rates in the regularized game (2) sublinear last-iterate convergence rates to exact Nash equilibrium in monotone games, which covers two-player zero-sum games as a special case. Moreover, the results of [1] holds under both full-information and noisy feedback. Thus some of results in the current paper is subsumed by [1] which also gives several weakness:
 (a) The current results does not provide convergecne rates to exact Nash equilibrium
 (b) The current results hold only for two-player zero-sum games but not the more general monotone games.
 (c) The current results hold only for full-information feedback.
2. By introducing regularization to the underlying two-player zero-sum game, the game becomes strongly monotone (strongly-convex-strongly-concave). Since RegFTRL is equivalent to running FTRL on a regularized strongly monotone game, the linear last-iterate convergence of RegFTRL follows form the fact that Mirror Descent (MD) or Follow the Regularized Leader (FTRL, the lazy projection version of MD) has linear last-iterate convergence. This approach is also studied in many recent works [1,2] and the paper should discuss the difference and their unique contribution more clearly.
3. " In practical terms, the implementation of the optimistic update approach often necessitates the computation of multiple gradients at each iteration, making it intricate and resource-intensive. " This is not true for OMWU or OGDA (refers to Optimistic Gradient Descent-Ascent) which only requires computation of one gradient in each iteration.
4. Some missing references on related works. Recent works [3, 4] have proved tight last-iterate convergecne rates of extragradient and OGDA *without* the unique Nash equilibrium assumption in monotone games. More recently, [5] proved last-iterate convergence rates in two-player zero-sum games (also without the unique Nash equilibrium assumption) with *bandit feedback* using *only* regularization. The result of [5] also shows that regularization is enough for last-iterate convergence rates for zero-sum games, with even more limited feedback.


[1] Slingshot Perturbation to Learning in Monotone Games. Abe et al., ArXiv 2023

[2] Last-Iterate Convergence with Full and Noisy Feedback in Two-Player Zero-Sum Games. Abe et al. AISTATS 2023

[3] Finite-Time Last-Iterate Convergence for Learning in Multi-Player Games. Cai et al., NeurIPS 2022

[4] Last-Iterate Convergence of Optimistic Gradient Method for Monotone Variational Inequalities, Gorbunov et al. NeurIPS 2022

[5] Uncoupled and Convergent Learning in Two-Player Zero-Sum Markov Games. Cai et al. NeurIPS 2023

**Questions:**

I would like to know if the current approach gives more general results: (a) extension to monotone games; (b) last-iterate convergence *rates* to Nash equilibrium; (c) convergence (*rates*) under noisy feedback / bandit feedback.

---

> ### Author Response · Authors · 2023-11-18
> **response to Reviewer 3dkM**
>
> Thanks for your comments which are quite helpful to further improve the quality of the paper.  We have made some necessary revisions (marked in blue font) in this version. Please refer to  the uploaded new version of submission and our point-to-piont responses below.
>
> **1. The proposed approach in this paper is very similar to the approach proposed in [1]**
>
> We thank the reviewer for pointing out work [1], and acknowledge our work shares some similarities with it. However, we would like to highlight some distinction between these two approaches.
>
> 1. Our paper focuses a complementary scenario. While [1] considers a broader class of monotone games beyond NFGs, their analysis could not encompass the behavior-form EFGs considered in our paper.
> 2. When concentrating on NFGs under full-information feedback, our analyses differ from [1], leading to distinct conclusions. Specifically:
>    - Our proposed step size range is larger, especially when $g=D_{\psi}$, where our range is $(0, \frac{1}{\tau}]$ compared to their $(0, \frac{2\tau}{3\tau^2+8})$.
>    - The derived convergence rates differ slightly. Our rates are of the form $(1+\eta\tau\lambda)^{-t}$, whereas [1] is $(1-\eta\tau\lambda/2)^{t}$.
> 3. Additionally, we have strengthened our results by adding sublinear rates to the exact Nash equilibrium, i.e, Theorem 4 in the revised version (see Theorem 4 on page 7). Notably, our results hold for any general-case regularization, whereas [1] specifically considers the l2-norm.
>
> To enhance clarity, we have added a comparison with the relevant literature, especially [1], in the related work (see Section 1.1 on page 3) and the remark following our theorems (see Remark 2 on page 6).
>
>
> **2. “In practical terms, the implementation of the optimistic update approach often necessitates the computation of multiple gradients at each iteration, making it intricate and resource-intensive. " This is not true for OMWU or OGDA (refers to Optimistic Gradient Descent-Ascent) which only requires computation of one gradient in each iteration.**
>
> Our original wording was indeed misleading. We have modified "computation of multiple gradients" to "computation of multiple strategies" to accurately convey the idea. In the context of RegFTRL, the computation involves a single strategy sequence $\pi_t$ at each iteration. On the other hand, in the case of OMWU or OGDA, maintenance of two strategy sequences, $\pi_t$ and $\hat\pi_t$, is required. The computation of $\pi_{t+1}$ in each iteration involves both $\pi_t$ and $\hat\pi_{t+1}$. Therefore, the distinction lies in the number of strategy sequences involved in the computation. Furthermore, OGDA has high per-iteration complexity due to the costly projection operations at each iteration, which adds to the computational burden.
>
>
> **3. Some missing references on related works.**
>
> 1. We acknowledge that references [3, 4] are relevant works utilizing optimistic/extra gradient techniques. In the revised version, we have included these references in the related works section to provide a more comprehensive overview of the literature (see Section 1.1 on page 2~3).
> 2. Regarding reference [5], we recognize the relevance of this work, especially in the context of entropy regularization. However, we want to emphasize the distinction between our work and the scenario considered in [5]:
>    - In stateless NFGs, while [5] focuses on bandit feedback in the context of entropy regularization, our work addresses full feedback scenarios under general-case regularization.
>    - In sequential decision making settings, the scope of analysis differs, with [5] exploring infinite-horizon discounted Markov games and path convergence, whereas our work concentrates on undiscounted behavior-form EFGs and last-iterate convergence. These differences in emphasis and setting make direct comparisons difficult, and the conclusions are not easily transferable between the two analyses.
>
>
> [1] Slingshot Perturbation to Learning in Monotone Games. Abe et al., ArXiv 2023
>
> [2] Last-Iterate Convergence with Full and Noisy Feedback in Two-Player Zero-Sum Games. Abe et al. AISTATS 2023
>
> [3] Finite-Time Last-Iterate Convergence for Learning in Multi-Player Games. Cai et al., NeurIPS 2022
>
> [4] Last-Iterate Convergence of Optimistic Gradient Method for Monotone Variational Inequalities, Gorbunov et al. NeurIPS 2022
>
> [5] Uncoupled and Convergent Learning in Two-Player Zero-Sum Markov Games. Cai et al. NeurIPS 2023

---

> > ### Comment · Reviewer_3dkM · 2023-11-22
> >
> > I thank the authors for their very detailed response. I have one question on Theorem 4.
> >
> > I think Theorem 4 only holds for l2-norm but you claim "Notably, our results hold for any general-case regularization, whereas [1] specifically considers the l2-norm". In my understanding, the proof of Theorem 4 relis on Lemma 3, which is a restatement of Lemma E.2 in [1]. However, Lemma E.2 in [1] only holds for l2-norm. Thus, it is confusing that the result of [1] holds only for l2-norm but Theorem 4 holds for any general-case regularization.
> >
> > [1] Slingshot Perturbation to Learning in Monotone Games. Abe et al., ArXiv 2023

---

> > > ### Author Response · Authors · 2023-11-22
> > > **response to Reviewer 3dkM**
> > >
> > > We would like to express our sincere apologies for the oversight on the revised version of Theorem 4. We have reverted to the original statement of Theorem 4, guaranteeing the asymptotic convergence to the Nash Equilibrium. However, we would like to emphasize that previous work typically provides asymptotic convergence only, whereas Abe's work [1] was conducted concurrently with ours. While we share some results with Abe's work, it is crucial to note that the analytical proof they employ differs from ours, and their results do not extend to the behavior-form EFGs in our paper. Moving forward,  we acknowledge the complexity associated with the extension of Theorem 4 to general-case regularization with guaranteed rate, which is also an interesting question in future research.

---

> > > > ### Author Response · Authors · 2023-11-22
> > > >
> > > > Given the time constraints, and assuming we have sufficiently addressed the concerns you raised, would you consider re-evaluating our score? We sincerely appreciate your assistance and understanding.

---

### Official Review · Reviewer_43MX · 2023-10-31

**Soundness:** 3 good
**Presentation:** 1 poor
**Contribution:** 3 good
**Rating:** 5
**Confidence:** 3

**Summary:**

This paper presents a last-iterate convergent algorithm for equilibrium computation in NFGs and EFGs that does not rely on optimism or uniqueness assumptions. In NFGs, it achieves convergence to an exact Nash equilibrium, whereas in EFGs it converges to a QRE. The paper also presents an implementation of the algorithm which utilizes neural network based function approximation, which is useful in large-scale settings. Finally, it presents numerical evidence to demonstrate the convergence of the framework of algorithms presented.

I thank the reviewers for their responses to my questions. I would like to maintain my score at this time.

**Strengths:**

1. The paper makes a solid technical contribution towards understanding last-iterate convergence in games. In particular, obviating the need for optimism and the uniqueness of equilibrium assumption is quite interesting.
2. The paper presents numerical simulations of the algorithmic framework, demonstrating fast convergence in NFGs and EFGs, likely competitive with the SOTA.

**Weaknesses:**

1. The paper requires significant proofreading. There are many typos and missing articles (e.g., "continue-time" and "continues-time" should be "continuous-time" on page 5) and quantities aren't necessarily always clearly defined (e.g., $r^h$ should be either explicitly given a name or otherwise introduced at the bottom of page 3 since the way it is currently written it is assume that the reader should know what $r^h$ is/that it already has been introduced).
2. The preliminaries could be made more substantial. A discussion of FTRL (at least explicitly mentioning the FTRL update) would be appropriate in the preliminaries (or earlier in the introduction).

**Questions:**

1. Perhaps you can note once after the preliminaries all proofs are included in the appendix instead of explicitly creating a proof environment for each theorem statement to state that the proof can be found in the appendix.
2. Have you considered mentioning the last-iterate analysis of OGDA that has been done by Wei et al. 2021 in the related work section?
3. Why do the plots in Figures 3 and 4 start at $10^2$ and $10^3$? It seems important to note the performance early on as well.
4. Is there a reason you are not comparing to SOTA CFR variants (e.g. CFR$^+$ [a], DCFR [b], PCFR$^+$ [c]) in your EFG experiments in Figure 4?

[a] Solving Large Imperfect Information Games Using CFR$^+$. Oskari Tammelin, 2014.
[b] Solving Imperfect-Information Games via Discounted Regret Minimization. Noam Brown and Tuomas Sandholm, 2019.
[c] Faster Game Solving via Predictive Blackwell Approachability: Connecting Regret Matching and Mirror Descent. Gabriele Farina, Christian Kroer, and Tuomas Sandholm, 2019.

---

> ### Author Response · Authors · 2023-11-18
> **response to Reviewer 43MX**
>
> Thanks for your comments which are quite helpful to further improve the quality of the paper.  We have made some necessary revisions (marked in blue font) in this version. Please refer to  the uploaded new version of submission and our point-to-piont responses below.
>
> **1. The paper requires significant proofreading. There are many typos and missing articles (e.g., "continue-time" and "continues-time" should be "continuous-time" on page 5) and quantities aren't necessarily always clearly defined (e.g., $r^h$ should be either explicitly given a name or otherwise introduced at the bottom of page 3 since the way it is currently written it is assume that the reader should know what $r^h$ is/that it already has been introduced).**
>
> We have proofread the paper several times again in order to eliminate any possible typos.
>
>
>
> **2. The preliminaries could be made more substantial. A discussion of FTRL (at least explicitly mentioning the FTRL update) would be appropriate in the preliminaries (or earlier in the introduction).**
>
> Following your advice, in the main text, we have added an explicit but brief mention of the FTRL update in the preliminaries (see Section 2.2 on page 4), and provide a detailed explanation in the Appendix C to adhere to page limitations.
>
>
>
> **3. Perhaps you can note once after the preliminaries all proofs are included in the appendix instead of explicitly creating a proof environment for each theorem statement to state that the proof can be found in the appendix.**
>
> Thanks for the advice, and we have adopted your recommendation in the revised version.
>
>
>
> **4. Have you considered mentioning the last-iterate analysis of OGDA that has been done by Wei et al. 2021 in the related work section?**
>
> While OGDA is indeed an important related work, we initially did not consider mentioning it in the related work section as its connection to our paper may not be as close as that of OMWU. Nevertheless, following your advice, we have included a reference to OGDA by Wei et al. 2021 in the revised version to provide a more comprehensive overview of the relevant literature (see Section 1.1 on page 2).
>
>
>
> **5. Why do the plots in Figures 3 and 4 start at $10^2$ and $10^3$? It seems important to note the performance early on as well.**
>
> In the early iterations, the exploitability of various algorithms experiences rapid declines, and the distinction among them is not apparent. By focusing on later iterations, where the differences among these algorithms become more evident, we aimed to provide a clearer and more meaningful comparison. However, we acknowledge the importance of presenting a complete picture. In response to your feedback, we have added figures in the revised paper to include all iterations (see Figure 3 and 4 on page 8), ensuring a more comprehensive view of the algorithms' performance over the entire range.
>
>
>
> **6. Is there a reason you are not comparing to SOTA CFR variants (e.g. CFR+ [a], DCFR [b], PCFR+ [c]) in your EFG experiments in Figure 4?**
>
> We have added CFR+ as a baseline in the revised version (see Figure 4 on page 8).

---

> > ### Author Response · Authors · 2023-11-22
> >
> > Given the time constraints, and assuming we have sufficiently addressed the concerns you raised, would you consider re-evaluating our score? We sincerely appreciate your assistance and understanding.

---

### Official Review · Reviewer_s7M3 · 2023-11-01

**Soundness:** 2 fair
**Presentation:** 2 fair
**Contribution:** 2 fair
**Rating:** 5
**Confidence:** 2

**Summary:**

The paper provides theoretical results for the regularized follow-the-regularized-leader (RegFTRL) algorithm, demonstrating last-iterate convergence in both normal-form and extensive-form games. It highlights a trade-off in the selection of the regularization parameter. Additionally, the authors propose two strategies: a gradual decrease of the regularization parameter and an adaptive adjustment of the reference strategy. Finally, the paper introduces an algorithm based on RegFTRL, extending its applicability to reinforcement learning.

**Strengths:**

The results presented in this paper have several advantages:

1. the guarantee is for the last iterate, which may make it more favorable in practice;

2. It does not need the uniqueness assumption that appears in other work.

**Weaknesses:**

**Novelty:**

The results presented are not surprising, as one might anticipate that incorporating regularization would enable the algorithm to achieve linear convergence in the last iteration. There are already existing works in the literature that demonstrate last-iterate convergence, such as [Wei et al., 2021], which diminishes the novelty of this result. The significance of the last-iterate result is also questionable, as it is just about the algorithm's output.

**Presentation:**

The preliminary section in Section 2 could benefit from clearer writing and more precise notation. For example, the notation $V^{h, \tau}$ is introduced in Section 2.1, but later, in Section 2.3, the paper uses $V^i$, with the superscript taking on a different meaning.

Additionally, the paper would be strengthened by an expanded discussion on certain results, such as after Theorem 2. A comparison with existing results regarding the convergence rate, given an appropriately chosen regularization parameter, would provide valuable context and insights.

**Rigor:**

Certain sections of the paper lack the necessary rigor. In Section 3.3, two approaches are presented: decreasing $\tau$ and changing reference strategies. However, the theoretical results in Theorems 2 and 3 are derived under the assumption of a constant $\tau$, and thus do not directly apply when $\tau$ is decreasing. A more rigorous approach would involve providing a specific sequence for $\tau$ and establishing the corresponding convergence rate, rather than merely stating that "The speed of convergence will be adversely affected as the weight parameter $\tau$ decreases," as is currently done. Regarding the changing reference strategy in Theorem 4, further clarification on the choice of $\tau$ and the associated rate would enhance the paper's comprehensiveness.

**Questions:**

While the value function Q is approximated using an actor-critic approach in equation (3), is it correct to assume that the Q function in equation (2) is known?

---

> ### Author Response · Authors · 2023-11-18
> **response to Reviewer s7M3**
>
> Thanks for your comments which are quite helpful to further improve the quality of the paper.  We have made some necessary revisions (marked in blue font) in this version. Please refer to  the uploaded new version of submission and our point-to-piont responses below.
>
>
> **1. Novelty**
>
> We acknowledge that there are existing works, such as [Wei et al., 2021], that explore the last-iterate convergence problem. However, it's worth noting that many of these works employ optimistic techniques or a combination of optimistic and regularization techniques. In contrast, our work emphasizes achieving last-iterate convergence solely through regularization techniques, providing a more straightforward and conceptually distinct approach.
>
> Besides, works such as [Sokota et al., 2023; Abe et al., 2022; Perolat et al., 2021], similar to ours, specifically focus on investigating the impact of regularization on last-iterate convergence. However, these works either focus on matrix games [Sokota et al., 2023; Abe et al., 2022] and only guarantee asymptotic convergence to the NE of the original game [Abe et al., 2022], or assume continuous-time feedback [Perolat et al., 2021].
>
> Our contribution lies in the exploration of the effects of general-case regularization on last-iterate convergence, providing further insights into discrete-time convergence rate to NE. Additionally, we extend our study to sequential decision-making settings, offering a comprehensive analysis.
>
> Although last-iterate convergence and average-iterate convergence only focus on the algorithm's output, their implications for algorithm performance are significant. For algorithms with only average-iterate convergence guarantees, theoretical considerations suggest that their last-iterate strategy may deviate from the optimal strategy or even diverge. In terms of implementation, dealing with average-iterate strategies poses challenges related to memory consumption and computational overhead.
>
> **2. Presentation**
>
> We appreciate your keen observation, and in response to your comment, we have made some revisions. Specifically, we have unified the notation to avoid ambiguity, addressing the inconsistency pointed out with $V^{h,\tau}$ and $V^i$. Additionally, we have expanded the discussion in the remarks following the theoretical results, providing a comparative analysis with relevant existing results (see Remark 2 on page 6).
>
>
> **3. Rigor**
>
> 1. Regarding the annealing approach, it is indeed inspired by the insights provided by Theorems 2 and 3: larger values of $\tau$ lead to faster convergence but introduce higher bias. While our experiments support its effectiveness, we recognize the current description may appear too absolute, and we have fixed it in the revised version (see Section 3.3 on page 6).
> 2. In the adaption approach, we appreciate your suggestion to provide a more detailed analysis of the convergence speed for different choices of $\tau$. In the revised version, we have introduced a strengthened Theorem 4 to illustrate the comparative convergence rates for various $\tau$  (see Theorem 4 on page 7). This theorem shows that both excessively large and small values of $\tau$ can slightly hinder the convergence rate. The explanation for this phenomenon can be derived from Theorems 2 and 3. Larger $\tau$ values result in faster convergence when the reference strategy $\mu_k$ is fixed, but at the cost of a larger bias, requiring more iterations to reduce it. On the other hand, smaller $\tau$ values, while having a smaller bias, lead to slower convergence, necessitating more iterations to converge. Thus, an optimal choice for $\tau$ lies in a moderate range.
>
> **4. While the value function Q is approximated using an actor-critic approach in equation (3), is it correct to assume that the Q function in equation (2) is known?**
>
> Yes, the assumption that the Q function in equation (2) is known corresponds to a full-information setting. This is a foundational assumption shared by many works in the literature, including [Sokota et al., 2023; Perolat et al., 2021; Abe et al., 2022].
>
> [Wei et al., 2021] Last-iterate Convergence of Decentralized Optimistic Gradient Descent/Ascent in Infinite-horizon Competitive Markov Games.
>
> [Sokota et al., 2023] A Unified Approach to Reinforcement Learning, Quantal Response Equilibria, and Two-Player Zero-Sum Games.
>
> [Perolat et al., 2021] From Poincaré Recurrence to Convergence in Imperfect Information Games: Finding Equilibrium via Regularization.
>
> [Abe et al., 2022] Last-Iterate Convergence with Full and Noisy Feedback in Two-Player Zero-Sum Games.

---

> > ### Author Response · Authors · 2023-11-22
> >
> > We would like to express our sincere apologies for the oversight on the revised version of Theorem 4 (please refer to https://openreview.net/forum?id=qjFnENGhDE&noteId=yVUfV1z1Da). We have reverted to the original statement of Theorem 4, guaranteeing the asymptotic convergence to the Nash Equilibrium.
> >
> > For the answer to adaption approach in "Q3 Rigor", we still maintain our perspective on the choice of parameter $\tau$ and promise to add sufficient ablation experiments to prove our argument in subsequent versions.

---

> > > ### Comment · Reviewer_s7M3 · 2023-11-22
> > >
> > > I would like to thank the authors for their response and the efforts they have made to enhance the paper.
> > >
> > > After reviewing the authors' responses to my comments and those of the other reviewer, I have a few comments. Concerning the significance of the last-iterate output, I believe that the "memory consumption and computational overhead" are quite minimal in this instance. However, I do concur that algorithms offering a last-iterate guarantee are likely more practical in application. As for the comparison with optimistic algorithms discussed in Reviewer 3dkM's comments, I think that the requirement to maintain two policies is not a significant disadvantage for optimistic algorithms. It's also worth noting that algorithms necessitating regularization can be adversely affected by an improper choice of regularization parameters.

---

> > > > ### Author Response · Authors · 2023-11-22
> > > > **response to Reviewer s7M3**
> > > >
> > > > Thanks for your comments.
> > > >
> > > > **(1) For the significance of the last-iterate convergence.**
> > > >
> > > > We still maintain our opinion that last-iterate convergence is an important property of game-theoretic algorithms, which is also supported by many studies:
> > > >
> > > > 1. (from page 2 in [1]) From a game-theoretic perspective, the last-iterate convergence is more appealing compared to the time-average convergence, as only the last-iterate convergence provides a description of the evolution of the overall behavior of the players.
> > > > 2. (from page 1 in [2]) Additionally, the averaging procedure can create complications. It not only increases the computational and memory overhead, but also makes things difficult when incorporating neural networks in the solution process, where averaging is usually not possible. Indeed, to address this issue, Brown et al. [2019] create a separate neural network to approximate the average strategy in their Deep CFR model.
> > > > 3. (from page 10 in [3]) Currently, the average-iterate convergence of CFR is an obstacle to using function approximation. In the seminal work Deep-CFR (Brown et al., 2019), the authors trained an additional network to maintain the average policy, which caused additional approximation errors. In the subsequent work (Steinberger, 2019; Steinberger et al., 2020), to get the average policy, they stored the networks at every iteration on disk and sampled one randomly to follow. Though sampling successfully eliminates the additional approximation error, given that it takes at least 10^5 iterations to converge in large poker games, storing all networks on disk is not tractable for large games like Texas Hold’em.
> > > >
> > > >
> > > > **(2) For the comparison with optimistic.**
> > > >
> > > > We first acknowledge that an improper choice of regularization parameters may lead to the performance degradation of the algorithm. However, based on Theorem 4 in our paper, there is no stipulated requirement on the regularization parameters for convergence to the Nash equilibrium. This suggests that our adaptation approach could be more robust to parameter choices.
> > > >
> > > > Regarding the optimistic update, it's important to note that some studies highlight certain drawbacks. For example, authors in [4] point out that extragradient or optimistic gradient methods typically require multiple gradient evaluations at each iteration and are more complicated to implement (see page 3 in [4]). Additionally, authors in [5] also point out that OGDA has a higtcomputational overhead (see page 3 in [5]). Furthermore, empirical evidence from literature [6] suggests that OMWU struggles to converge in noisy-information feedback settings, while M2WU (an instantiation of RegFTRL with moment projection) converges rapidly.
> > > >
> > > > Therefore, we posit that regularization-based methods represent a valuable class, complementing the optimistic update approach.
> > > >
> > > >
> > > > [1] Cai et al., Finite-time last-Iierate convergence for learning in multi-player games.
> > > >
> > > > [2] Lee et al., Last-iterate convergence in extensive-form games.
> > > >
> > > > [3] Liu et al., The power of regularization in solving extensive-form games.
> > > >
> > > > [4] Zeng et al., Regularized gradient descent ascent for two-player zero-sum Markov games.
> > > >
> > > > [5] Meng et al., Efficient last-iterate convergence algorithms in solving games.
> > > >
> > > > [6] Abe et al., Last-iterate convergence with full-and noisy-information feedback in two-player zero-sum games.

---

> > > > > ### Author Response · Authors · 2023-11-22
> > > > >
> > > > > Given the time constraints, and assuming we have sufficiently addressed the concerns you raised, would you consider re-evaluating our score? We sincerely appreciate your assistance and understanding.

---

### Official Review · Reviewer_gsq4 · 2023-11-05

**Soundness:** 2 fair
**Presentation:** 2 fair
**Contribution:** 4 excellent
**Rating:** 8
**Confidence:** 5

**Summary:**

This study introduces Regularized Follow-the-Regularized-Leader (RegFTRL), an innovative method for equilibrium learning in two-player zero-sum games. RegFTRL, an improved form of FTRL, incorporates a dynamic regularization mechanism that includes the familiar entropy regularization. Within normal-form games (NFGs), RegFTRL demonstrates a promising quality of swift, linear convergence to an estimated equilibrium and can adjust to achieve exact Nash equilibrium. When applied to extensive-form games (EFGs), the entropy-regularized version of RegFTRL, specifically through the Multiplicative Weights Update (MWU) technique, also attains linear convergence to the quantal response equilibrium without depending on optimistic updates or unique conditions. This illustrates that regularization alone can ensure direct convergence. The paper also presents FollowMu, an applied variant of RegFTRL using neural networks for function approximation in learning within evolving sequential settings. Empirical evidence confirms RegFTRL's theoretical advantages and shows that FollowMu performs well in EFGs.

**Strengths:**

I can succinctly state that a high-quality paper's value is self-evident and does not require elaborate explanation. Regarding zero-sum games, exponential convergence has already been established by seminal works such as Wei et al. (ICLR 2020) and Panageas (NeurIPS 2019). However, this paper presents a method characterized by its simplicity of proof and seamless application to extensive-form games (EFGs). On the basis of its theoretical contributions, this is the main reason I view the paper favorably.

**Weaknesses:**

Part of the results have been already proposed in the literature via different analysis.
I think that authors already understood that their presentation could be improved especially in presenting of the algorithm
but I understand that it is due to the page limits

**Questions:**

I did not have the time to delve in the details of the proof due to the always pressing schedule of ICLR but I would like to ask some questions to be sure that I understand correctly the result:

1) Do you request uniqueness of NE in zero-sum game?
2) In NFGs (0-sum), your algorithm converge always to an \eps-NE?
3) What is the reason that you pass to FollowMu in the experimental section?

---

> ### Author Response · Authors · 2023-11-18
> **response to Reviewer gsq4**
>
> Thanks for your comments which are quite helpful to further improve the quality of the paper.  We have made some necessary revisions (marked in blue font) in this version. Please refer to  the uploaded new version of submission and our point-to-piont responses below.
>
> **1. Do you request uniqueness of NE in zero-sum game?**
>
> No, we do not request the uniqueness of NE. The uniqueness of the equilibrium is ensured by Theorem 1 in our paper, and therefore, we do not rely on the uniqueness assumption.
>
>
>
> **2. In NFGs (0-sum), your algorithm converge always to an \eps-NE?**
>
> Yes, our algorithm converges to an $\epsilon$-NE in NFGs without the use of annealing or adaption approaches. However, as indicated by experiments and theorems in our paper, the proposed algorithm can also converge to an NE through the adaption approach.
>
>
>
> **3. What is the reason that you pass to FollowMu in the experimental section?**
>
> In the experimental section, we employed FollowMu as part of a two-fold evaluation. The first evaluation assesses RegFTRL to substantiate our theoretical findings, while the second one evaluates the neural-based algorithm, FollowMu, which is derived from RegFTRL, in order to validate the effectiveness and practical performance of RegFTRL with neural network function approximation.

---

### Author Response · Authors · 2023-11-21

We have attempted to address the reviewers' questions. If there are any points in our responses may not fully address your concerns, please let us know and we will get back to you as soon as possible.

---

### Public Comment · ~Mingyang_Liu1 · 2023-11-23
**About the notation**

Dear authors,

I like this paper very much since it provides an intriguing and novel perspective on last-iterate convergence in Extensive-Form Games (EFGs) through an actor-critic approach. This is a commendable effort and certainly adds value to the field.

While delving into the technical aspects, particularly the proof sections, I encountered a point of confusion regarding the reward function $r^h(I, a)$. Specifically, in the first line of Lemma 10, it appears that the $r^h(I,a)$ for both $Q_{t_2}^{h-1,\tau}$ and $Q_{\pi_\mu}^{h-1,\tau}$ cancel each other out. However, based on the provided definition where $r^h(I,a)=\sum_{s\in (I,a)}\rho^\pi(s,a) r^h(s,a) / \sum_{s\in I} \rho^\pi(s)$, there seems to be a dependency on the current strategy of $\pi$. This leads me to question if they would indeed cancel out in $Q_{t_2}^{h-1,\tau}$ and $Q_{\pi_\mu}^{h-1,\tau}$.

I'm sure I might be overlooking a detail or misinterpreting a part of the proof. Could you kindly help clarify this for me? I am eager to fully grasp the nuances of your methodology and appreciate any guidance you can provide to help me understand this better.

Thank you for your time and for sharing your valuable work.

---

### Meta-Review · Area_Chair_tDzP · 2023-12-07

**Metareview:**

The paper introduces a (continuous time) variant of FTRL by adding a new regularization term in the value function of a two player zero-sum game. It is established that the modified game has a unique NE equilibrium and as a result it leads to last iterate convergence guarantees. The paper has interesting results though are not surprising and follow the idea of entropic regularization that has extensively appeared in the literature for QRE (Quantal response NE) applied for both continuous and discrete dynamics. Most of the reviewers believe that the paper is slightly below the bar for ICLR, as there is not enough technical novelty.

**Justification For Why Not Higher Score:**

The paper seems to lack technical novelty and received borderline scores.

**Justification For Why Not Lower Score:**

NA

---

### Decision · Program_Chairs · 2024-01-16

Reject